# Risk-Averse Constrained Reinforcement Learning with Optimized Certainty Equivalents

**Jane H. Lee**
Department of Computer Science
Yale University
jane.h.lee@yale.edu

**Baturay Saglam**
Department of Electrical Engineering
Yale University
baturay.saglam@yale.edu

**Spyridon Pougkakiotis**
Department of Mathematics
King's College London
spyridon.pougkakiotis@kcl.ac.uk

**Amin Karbasi**[*]
Foundation AI
Cisco Systems Inc.
karbasi@cisco.com

**Dionysis Kalogerias**
Department of Electrical Engineering
Yale University
dionysis.kalogerias@yale.edu

## Abstract

Constrained optimization provides a common framework for dealing with conflicting objectives in reinforcement learning (RL). In most of these settings, the objectives (and constraints) are expressed though the expected accumulated reward. However, this formulation neglects risky or even possibly catastrophic events at the tails of the reward distribution, and is often insufficient for high-stakes applications in which the risk involved in outliers is critical. In this work, we propose a framework for risk-aware constrained RL, which exhibits per-stage robustness properties jointly in reward values and time using optimized certainty equivalents (OCEs). Our framework ensures an exact equivalent to the original constrained problem within a parameterized strong Lagrangian duality framework under appropriate constraint qualifications, and yields a simple algorithmic recipe which can be wrapped around standard RL solvers, such as PPO. Lastly, we establish the convergence of the proposed algorithm under common assumptions, and verify the risk-aware properties of our approach through several numerical experiments.

## 1 Introduction

Autonomous agents are often used in settings where they must handle several conflicting requirements while maximizing a main objective, such as navigating a maze without hitting any walls. These conflicting requirements are often modeled with constrained reinforcement learning (RL) (see Sutton and Barto [42]) and are either solved as a multi-objective problem with conflicting requirements controlled by weights, chosen manually or through hyperparameter tuning (e.g., see Borkar [10], Achiam et al. [2], Mannor and Shimkin [28], Tessler et al. [45], Mania et al. [27]), or by using primal-dual methods which optimize over the weights [45, 5]. Others have approached constrained RL through reward-free techniques, e.g., [30], or strategic exploration [54]. The work of Paternain et al. [33] has given theoretical support to primal-dual approaches by proving that, despite their

---

[*]Work done while affiliated with Yale University.

39th Conference on Neural Information Processing Systems (NeurIPS 2025).

inherent nonconvexity, constrained RL models exhibit zero duality gap under mild assumptions. However, in the standard (risk-neutral) setting (including that of Paternain et al. [33]), the objectives and constraints are often formulated as expected values of a (discounted) sum of rewards. Nonetheless, in many practical scenarios, especially in the context of safety [24] and other high-stakes applications, the standard expectation may not be strict enough to describe the desired behavior. For example, instead of maximizing the on-average return (e.g., of an investment), one may want to learn policies that mitigate risk (e.g., of losing money).

**Stochastic Optimization with Risk Measures** Risk measures are often used to quantify this risk in problems with uncertain (random) outcomes (see, e.g., Shapiro et al. [41]). More broadly, stochastic optimization with risk measures in the objectives and/or constraints has been studied in the online setting [26], bandits [11, 37, 44], statistical learning [3, 15, 32, 48], non-convex resource allocation problems [21], and others [20, 55]. These methods and analyses are not readily applicable in our setting either due to non-convexity/concavity in the objective and constraints or due to the fact that the risk envelope for our problem has dependence on the policy $\pi$.

**Risk-Averse/Risk-Aware RL** In reinforcement learning specifically, there is a deep literature on risk-averse methods [31, 53, 13, 14, 29, 35, 49, 51, 50] with many applications in the area of robust control [52, 23]. While the most conservative approach consists of learning policies robust to worst-case scenarios [1, 18], we often prefer to take into account a set of events with significant probability. To do this, Huang et al. [17], Tamar et al. [43] consider the policy gradient method under coherent risk measures, where the risk measure replaces the expectation over the discounted sum of rewards. The work of Chow et al. [12] handles similar CVaR risk objectives with CVaR risk constraints by utilizing Lagrangian relaxation. Xu et al. [51], Wang et al. [50] study risk-sensitive RL with OCEs. Recently, Bonetti et al. [7] propose an alternative risk-aware framework for (unconstrained) reinforcement learning which places the risk measure in the occupancy measure to derive what the authors call the reward-based conditional value at risk (RCVaR) and mean-mean absolute deviation (Mean-RMAD), in contrast to the standard return-based formulations (like that of [12, 17, 43]).

**Contributions** This work approaches risk-aware constrained RL through employing reward-based risk measures in both the objective and the constraints. Our contributions are as follows:

- We extend the setting of Bonetti et al. [7] and, to the best of our knowledge, our work is the first to handle reward-based constraints, covering a large class of risk measures (OCEs);
- We establish a parameterized strong duality relation resulting in a computationally tractable partial Lagrangian relaxation (which is exact under certain constraint qualifications);
- We propose an online algorithm which is modular, offering the user flexibility in modeling (allowing for a mix of risk-neutral or risk-averse objectives and/or constraints) and implementation (enabling the use of existing RL algorithms as a black-box). By rigorously reducing the underlying problem to a certain instance of stochastic minimax optimization, we then establish convergence under common assumptions.
- We demonstrate practical usefulness of our approach in extensive numerical experiments on standard benchmarks, showcasing its effectiveness in reducing risk constraint violations and improving stochastic stability through explicit risk management.[2]

The setting of Bonetti et al. [7] is closely related to this work but is restricted to the unconstrained case, while our setting handles reward-based risk constraints as well. Our algorithm differs from the methodology of [7], which is based on a block-coordinate descent scheme, the analysis of which does not readily extend to the type of optimization problems arising in the constrained setting. Further, [7] requires exact policy solvers but we work with inexact ones (cf. Assumption 3.6). Compared to other risk-averse constrained RL methods, our partial Lagrangian relaxation is exact under a certain constraint qualification, whereas no meaningful (let alone exact) relation between the original primal problem and the employed Lagrangian relaxation has been shown in other works (e.g., [12]).

## 2 Background

**Notation** Given $m \in \mathbb{N}$, we let $[m] := \{1, \ldots, m\}$. Given $x \in \mathbb{R}$, we write $(x)_+ \equiv \max\{x, 0\}$. We denote by $\mathcal{P}(\mathcal{S})$ the space of probability distributions over $\mathcal{S}$, equipped with its Borel $\sigma$-algebra.

---

[2]We provide the code at `https://github.com/baturaysaglam/risk-averse-constrained-RL`.

## 2.1 Risk Measures

In the context of stochastic optimization, is it well-known that expectations are unable to capture "risky" events (related to statistical variability, dispersion, or fat-tail behavior of the randomness). *Risk-averse optimization* aims to minimize the risk associated with such events, which is captured by certain functionals known as *risk measures*. For an extended discussion on risk-averse optimization, we refer the reader to Shapiro et al. [41, Chapter 6].

**Optimized Certainty Equivalents (OCEs)** Among different classes of risk measures, in this work we are interested in those which are in *infimal convolution* form, i.e., $\rho(Z) = \inf_t\{t + \mathbb{E}\left[g(Z - t)\right]\}$, assuming that $Z$ represents losses. A notable example is the so-called *Conditional Value-at-Risk* at level $\beta$ (denoted by $\mathrm{CVaR}^\beta(Z)$), which is retrieved by setting $g(\cdot) = \frac{1}{\beta}(\cdot)_+$, for some $\beta \in (0, 1)$.

If $Z$ represents rewards (as often in RL), we consider risk measures in the *supremal convolution* form, i.e., $\widetilde{\rho}(Z) = \sup_t\{t + \mathbb{E}\left[g(Z - t)\right]\}$. For example, we can consider the reflected $-\mathrm{CVaR}^\beta(-Z)$ (for reward maximization) by setting $g(u) = -\frac{1}{\beta}(-u)_+$. In this notation, $g$ is called a *utility function*. Under certain conditions on the utility (e.g., satisfied by $g(u) = -(-u)_+$), these risk measures are known as *optimized certainty equivalents* (OCEs) (see Ben-Tal and Teboulle [4, Definition 2.2], where the reader can find many practical examples and properties). The class of OCEs is very rich (including many risk measures of practical value) and satisfies several important properties.

## 2.2 Risk-Neutral RL

Let $\tau \in \mathbb{N} \cup \{0\}$ denote the time instant and let $\mathcal{S} \subset \mathbb{R}^n$ and $\mathcal{A} \subset \mathbb{R}^d$ be compact sets describing the possible states and actions of an agent. Reinforcement learning is often modeled as a Markovian dynamical system where given a state and action pair at time $\tau$, say $(s_\tau, a_\tau)$, the next state distribution only depends on the current state-action pair (and is independent of $(s_{<\tau}, a_{<\tau})$). The agent chooses actions at each time instant using a policy $\pi \in \mathcal{P}(\mathcal{S})$, and the action taken by the agent results in rewards defined by the uniformly bounded function $r : \mathcal{S} \times \mathcal{A} \to \mathbb{R}$. We next describe the standard RL objective, that is, to maximize the (discounted, infinite) expected sum of rewards.

*Problem* (**Primal RL Formulation**).

$$V^* := \sup_{\pi \in \mathcal{P}(\mathcal{S})} \left\{ V(\pi) := \mathbb{E}\left[\sum_{\tau=0}^{\infty} \gamma^\tau r(s_\tau^\pi, a_\tau^\pi)\right]\right\}, \tag{1}$$

where $(s_\tau^\pi, a_\tau^\pi)$ denotes the state-action vector evolving under policy $\pi$. It is well-known that the above can be reformulated in terms of the (discounted) *occupancy (or occupation) measure* of a policy, defined as the discounted mixture $\mathrm{d}\nu^\pi(\cdot, \bullet) = (1 - \gamma)\sum_{\tau=0}^{\infty} \gamma^\tau \mathrm{d}p_\pi^\tau(\cdot, \bullet)$, where $p_\pi^\tau$ denotes the Borel probability measure induced by vector $(s_\tau^\pi, a_\tau^\pi)$. Let $\mathcal{R}$ denote the convex and weakly compact set of all occupation measures [8] induced by the policies $\pi \in \mathcal{P}(\mathcal{S})$. We can then rewrite the RL problem as follows.

*Problem* (**Convex Analytic Dual Form**[3]).

$$V^* := \sup_{\nu^\pi \in \mathcal{R}} \left\{ V(\nu^\pi) := \frac{1}{1 - \gamma} \cdot \mathbb{E}_{\nu^\pi}[r(s, a)]\right\}. \tag{2}$$

It is known that the occupation measure $\nu^\pi$ is in a one-to-one relationship with a policy $\pi$, i.e., if two policies have the same occupation measure, they must be the same (Sutton and Barto [42]).

## 2.3 Risk-Averse RL Formulations

There are several different frameworks for modeling "risk-awareness" and "safety" in RL. Existing works which explore risk-averse RL formulations have often taken the approach of replacing the

---

[3]This notion of casting dynamic optimization problems into abstract "static" optimization problems over a closed convex set of measures (as above) is referred to as the *convex analytic* approach by Borkar [9]. Thus, we will refer to these equivalent problems (1) and (2) as the "primal" and "convex analytic dual" formulations (not to be confused with duality in the Lagrangian sense) throughout.

Table 1: Comparison of the reward-based objective and the standard RL objective for general OCEs.

| FORMULATION | RISK-AVERSE (REWARD-BASED) | RISK-NEUTRAL |
|---|---|---|
| PRIMAL | $\displaystyle \sup_{\pi,t} \mathbb{E}\left[\sum_{\tau=0}^{\infty}\gamma^{\tau}\left(t+g(r(s_\tau^\pi,a_\tau^\pi)-t)\right)\right]$ | $\displaystyle \sup_{\pi} \mathbb{E}\left[\sum_{\tau=0}^{\infty}\gamma^{\tau}r(s_\tau^\pi,a_\tau^\pi)\right]$ |
| DUAL | $\displaystyle \sup_{\nu^\pi,t}\frac{1}{1-\gamma}\mathbb{E}_{\nu^\pi}\left[t+g(r(s,a)-t)\right]$ | $\displaystyle \sup_{\nu^\pi}\frac{1}{1-\gamma}\mathbb{E}_{\nu^\pi}\left[r(s,a)\right]$ |

expectations in the "primal" formulation (1) with risk measures, such as in the studies of Huang et al. [17], Tamar et al. [43], Chow et al. [12], among others:

$$\sup_{\pi\in\mathcal{P}(\mathcal{S})} -\rho\left[-\sum_{\tau=0}^{\infty}\gamma^{\tau}r(s_\tau^\pi,a_\tau^\pi)\right].\tag{3}$$

Alternatively, one could generalize the tower property of expectations and iteratively evaluate a risk measure at each decision stage (e.g., see Ruszczyński [36]). A less common formulation –of interest herein– *applies a risk measure on the occupancy measure*, rather than the original probability space, as in Bonetti et al. [7]:

$$R^* = \sup_{\nu^\pi\in\mathcal{R}}\frac{1}{1-\gamma}\cdot -\rho_{\nu^\pi(s,a)}(-r(s,a)),\tag{4}$$

where $\rho : \mathcal{L}_1(\nu^\pi,\mathbb{R})\to\mathbb{R}$ is a finite-valued risk measure. This choice (4) enforces robustness over both space and time, capturing *per-stage risk* in contrast to (3); for details see Appendix A.1.

**OCE Formulation**   By utilizing the relationship between (1) and (2) for risk measures induced by OCEs with a utility $g$, given that the reward is bounded, we obtain[4]

$$\frac{1}{1-\gamma}\sup_{\nu^\pi}\sup_{t}\mathbb{E}_{\nu^\pi}\left[t+g(r(s,a)-t)\right] \iff \sup_{t}\sup_{\pi}\mathbb{E}\left[\sum_{\tau=0}^{\infty}\gamma^{\tau}\left(t+g(r(s_\tau^\pi,a_\tau^\pi)-t)\right)\right].$$

This relationship is summarized in Table 1. As one can see, the risk-averse problem involves a maximization over an additional variable $t$ which depends on $\pi$ (in this sequence). Importantly, given a fixed $t$, maximization over $\pi$ resembles a problem of the form of (1) with a modified reward function. Thus one can take advantage of several existing algorithms which solve (1). This has been explored in the unconstrained setting of Bonetti et al. [7, Algorithm 2].

## 2.4   Constraints

Given the discussion in the previous section for risk-aware objectives, we can similarly describe constraints also of this form. It is common to address solving constrained problems through Lagrangian relaxation, which is amenable to numerical optimization, but (generally) only offers an approximate solution to the original constrained problem. We nonetheless show that the (partial) Lagrangian relaxation employed in this paper is *exact* under certain constraint qualifications.

Here, we highlight that Bonetti et al. [7] do *not* consider the constrained RL setting (which requires a dedicated analysis of the relation between the Lagrangian relaxation and the original constrained problem), while others using return-based risk objectives and constraints in the convex analytic "primal" space (e.g., Chow et al. [12]) are not able to show any meaningful relation between the primal problem and their employed Lagrangian relaxation.

## 3   Main Results (Risk Constrained Learning)

The framework we propose has favorable robustness properties, better capturing per-stage risk (see Appendix A.1) for applications which are also time-sensitive. Now, we: (1) show that this formulation

---

[4]Going forward, the expectation or risk without subscripts is taken over the initial state and sample path distribution induced by the state transitions of the system and the policy $\pi$.

allows us to solve the original constrained problem through a partial Lagrangian relaxation which is practically implementable and (2) reduce the problem to an instance of stochastic minimax optimization which lends itself to non-asymptotic convergence analysis.

## 3.1 Problem Description and Equivalent Reformulations

To avoid generalities and to facilitate exposition, hereafter we exclusively consider the $\mathrm{CVaR}^{\beta}$. However, all results presented hold for general OCEs. We also use the same $\beta$ for all constraints and the objective, but these need not be the same. We first write a constrained version of the problem (4):

$$\sup_{\nu^{\pi} \in \mathcal{R}} \frac{1}{1-\gamma} \cdot -\mathrm{CVaR}^{\beta}_{\nu^{\pi}}(-r_0(s,a)) \quad \text{s.t.} \quad \frac{1}{1-\gamma} \cdot -\mathrm{CVaR}^{\beta}_{\nu^{\pi}}(-r_i(s,a)) \geq c_i, \ \forall i \in [m], \quad (5)$$

or, equivalently, as:

$$\sup_{\pi \in \mathcal{P}(\mathcal{S})} \sup_{t_0 \in \mathbb{R}} \mathbb{E}\left[\sum_{\tau=0}^{\infty} \gamma^{\tau} \left(r_0'(s_\tau^\pi, a_\tau^\pi, t_0)\right)\right] \quad \text{s.t.} \quad \sup_{t_i \in \mathbb{R}} \mathbb{E}\left[\sum_{\tau=0}^{\infty} \gamma^{\tau} \left(r_i'(s_\tau^\pi, a_\tau^\pi, t_i)\right)\right] \geq c_i, \ \forall i \in [m], \quad (6)$$

where $r_i'(s,a,t) := t - \frac{1}{\beta}(t - r_i(s,a))_+$, and $r_i : \mathcal{S} \times \mathcal{A} \to \mathbb{R}$ are the reward functions for $i \in \{0\} \cup [m]$. Note that $r_i'$ is written specifically for the $\mathrm{CVaR}^{\beta}$ here, but for general OCEs we simply replace $-\frac{1}{\beta}(-\cdot)_+$ with $g(\cdot)$. Recall that, by standard convention, $r_i$ is bounded for all $i$.

*Remark* 3.1. Consider any OCE, and assume that $r_i$ are bounded for all $i \in \{0\} \cup [m]$. Then, by Ben-Tal and Teboulle [4, Proposition 2.1], we have that

$$\sup_{t \in \mathbb{R}} \mathbb{E}\left[\sum_{\tau=0}^{\infty} \gamma^{\tau} \left(t + g(r_i(s_\tau^\pi, a_\tau^\pi) - t)\right)\right] = \sup_{t \in \mathcal{T}_i} \mathbb{E}\left[\sum_{\tau=0}^{\infty} \gamma^{\tau} \left(t + g(r_i(s_\tau^\pi, a_\tau^\pi) - t)\right)\right],$$

where $\mathcal{T}_i$ is a bounded interval containing the uniformly smallest and largest values of the reward $r_i(s_\tau, a_\tau)$. Thus, we can (equivalently) cast problem (6), by restricting $t := [t_0, t_1, \ldots, t_m] \in \mathcal{T}$, with $\mathcal{T} \subset \mathbb{R}^{m+1}$ some convex and compact set. This will be utilized throughout.

**Lemma 3.2.** *Problem* (6) *is equivalent to*

$$\sup_{\substack{\pi \in \mathcal{P}(\mathcal{S}) \\ t \in \mathcal{T}}} \mathbb{E}\left[\sum_{\tau=0}^{\infty} \gamma^{\tau} \left(r_0'(s_\tau^\pi, a_\tau^\pi, t_0)\right)\right] \quad \text{s.t.} \quad \mathbb{E}\left[\sum_{\tau=0}^{\infty} \gamma^{\tau} \left(r_i'(s_\tau^\pi, a_\tau^\pi, t_i)\right)\right] \geq c_i, \ \forall i \in [m]. \quad (7)$$

The proof is based on Chow et al. [12], and given in Appendix A.2. Although it may seem somewhat redundant, (7) is a maximization problem jointly over the variables $\pi, t$, subject to functional inequality constraints. Thus, the Lagrangian associated to (7) reads $\mathcal{L}(\pi, t, \lambda) := \mathbb{E}[\widehat{\mathcal{L}}(\pi, t, \lambda)]$, with

$$\widehat{\mathcal{L}}(\pi, t, \lambda) := \sum_{\tau=0}^{\infty} \gamma^{\tau} \left[ \left(r_0'(s_\tau^\pi, a_\tau^\pi, t_0)\right) - \sum_{i=1}^{m} \lambda_i c_i + \sum_{i=1}^{m} \lambda_i \left(r_i'(s_\tau^\pi, a_\tau^\pi, t_i)\right) \right]. \quad (8)$$

Then, the primal problem is $P^* = \sup_{\pi,t} \inf_{\lambda \geq 0} \mathcal{L}(\pi, t, \lambda)$. Note that the above, for fixed $t, \lambda$, is the Lagrangian for a risk-neutral constrained problem (similar to those considered by Paternain et al. [33]), parameterized by $t$, which controls risk-aversion (in harmony with the choice of $\beta$).

## 3.2 Partial Lagrangian Relaxation

Next, we derive a partial Lagrangian relaxation for (7) (which is exact under a constraint qualification; cf. Assumption 3.4). In turn, this allows us to cast the problem in a stochastic minimax optimization framework and derive an effective online algorithm for its solutions (cf. Section 3.3).

**Partial Maximization Problem** Let us now consider problem (7) for fixed $t \in \mathcal{T}$, i.e.,

$$\sup_{\pi \in \mathcal{P}(\mathcal{S})} \mathbb{E}\left[\sum_{\tau=0}^{\infty} \gamma^{\tau} \left(r_0'(s_\tau^\pi, a_\tau^\pi, t_0)\right)\right] \quad \text{s.t.} \quad \mathbb{E}\left[\sum_{\tau=0}^{\infty} \gamma^{\tau} \left(r_i'(s_\tau^\pi, a_\tau^\pi, t_i)\right)\right] \geq c_i, \ \forall i \in [m]. \quad (9)$$

This problem exhibits strong duality under usual conditions.

**Proposition 3.3.** *Let $r_i$ be bounded functions for all $i \in \{0\} \cup [m]$. Assume that Slater's condition[5] holds for* (9)*. Then,* (9) *exhibits strong duality, and thus*

$$\sup_{\pi} \inf_{\lambda \geq 0} \mathcal{L}(\pi, t, \lambda) = \inf_{\lambda \geq 0} \sup_{\pi} \mathcal{L}(\pi, t, \lambda).$$

The proof of Proposition 3.3 is given in Appendix A.3.

**Maximizing over** $t$    From Remark 3.1, we know that $t \in \mathcal{T}$ and $\mathcal{T}$ is a convex and compact set (since we assume bounded rewards). Note that Proposition 3.3 holds for each fixed $t \in \mathcal{T}$, assuming Slater's condition is satisfied at this point. To proceed, it suffices to introduce another constraint qualification, as follows.

**Assumption 3.4** (Constraint Qualification). Let $t^*$ be an argument which attains the maximum over the variable $t = [t_0, t_1, \ldots, t_m]$ in (7). There is a (non-singleton) convex and compact set $\mathcal{I} \subset \mathcal{T}$, with $t^* \in \mathcal{I}$, such that Slater's condition holds for (9), for every $t \in \mathcal{I}$.

Under Assumption 3.4, we have that

$$\sup_{t \in \mathcal{I}} \sup_{\pi} \inf_{\lambda \geq 0} \mathcal{L}(\pi, t, \lambda) = \sup_{t \in \mathcal{I}} \inf_{\lambda \geq 0} \sup_{\pi} \mathcal{L}(\pi, t, \lambda).$$

**Parametrized Policies and Almost-zero Duality Gap**    In order to solve the (inner) policy optimization problem, we parametrize the policy by a vector $\theta \in \Theta \subset \mathbb{R}^p$, representing the coefficients of, say, a neural network, assuming that $\Theta$ is a compact set (which is the case in practice). We have shown that under Assumption 3.4, the proposed partial Lagrangian relaxation is exact. We argue that this remains almost true (in the sense described below) even if we utilize parametrized policies. In what follows, we assume that $\pi_\theta$ is an $\epsilon$-universal parametrization of measures in $\mathcal{P}(\mathcal{S})$, according to Paternain et al. [33, Definition 1].

**Theorem 3.5** (Almost-zero Duality Gap). *Let Assumption* 3.4 *hold, assume that $\pi_\theta$ is $\epsilon$-universal for measures in $\mathcal{P}(\mathcal{S})$, and that the policy-parametrized version of* (7) *is feasible. Then,*

$$P^*(t^*) := \sup_{t \in \mathcal{I}} \sup_{\pi} \inf_{\lambda \geq 0} \mathcal{L}(\pi, t, \lambda) \; \geq \; \sup_{t \in \mathcal{I}} \inf_{\lambda \geq 0} \sup_{\theta} \mathcal{L}(\pi_\theta, t, \lambda) \; \geq \; P^*(t^*) - \mathcal{O}\left(\frac{\epsilon}{1 - \gamma}\right).$$

The proof is given in Appendix A.4. Theorem 3.5 states that solving the parameterized (partial) dual problem incurs negligible cost (depending on the universal parametrization error $\epsilon > 0$) under mild assumptions and our usual constraint qualification (see Assumption 3.4).

**The Practical Parametrized Partial Dual Problem**    In practice, we do not have access to the set $\mathcal{I}$ (assuming that Assumption 3.4 is satisfied), and thus, we may heuristically search over all $t \in \mathcal{T}$, noting that a reasonable initial guess can maximize our chances of landing in $\mathcal{I}$ (and thus solving the original constrained problem (7) (almost) exactly). Nonetheless, the proposed algorithmic framework does not rely on Assumption 3.4. Indeed, in the absence of this, we recover a partial Lagrangian relaxation which yields an approximate solution to problem (7). However, we highlight that, unlike Lagrangian relaxations considered in the literature, our approach is exact under Assumption 3.4.

Additionally, from Proposition 3.3, we know that the optimal Lagrange multiplier is attained, assuming that Slater's condition holds for (7). Thus, there is an optimal Lagrange multiplier associated to (7) within some sufficiently large convex and compact set $\Lambda \subset \mathbb{R}_+^m$. In practice, the size of this set can be adjusted dynamically, if necessary. Hence, in what follows, we restrict the minimization over $\lambda \in \Lambda$. In light of the previous discussion, we focus on the following parametrized partial dual formulation:

$$D_\theta^* := \sup_{t \in \mathcal{T}} \inf_{\lambda \in \Lambda} \underbrace{\sup_{\theta} \mathcal{L}(\pi_\theta, t, \lambda)}_{\text{blackbox RL}}. \tag{10}$$

**A Wrapper for Blackbox Unconstrained RL Algorithms**    Notice that $\mathcal{L}(\pi_\theta, t, \lambda)$ is a linear combination of the expected discounted sum of adjusted reward functions which take the original $r_i$ to $r_i'(s, a, t_i) = t_i - \frac{1}{\beta}(t_i - r_i(s, a))_+$ for $i = 0, 1, \ldots, m$. Thus, the innermost problem in

---

[5]Slater's condition requires that there exists a strictly feasible policy.

---

**Algorithm 1** Reward-Based SGDA with Risk Constraints

---

1: **Input:** Bounded reward functions $r_i : \mathcal{S} \times \mathcal{A} \mapsto \mathbb{R}$, discount factor $\gamma \in (0, 1)$, step sizes $\eta_\lambda, \eta_t > 0$, batch size $n$.
2: **Initialize:** $t^{(0)}, \lambda^{(0)}$
3: **for** each iteration $j = 1, 2, \ldots, J$ **do**
4:     Call the inexact oracle to obtain $\pi_{\theta^{(j)}}$ satisfying Assumption 3.6.
5:     Collect a batch of trajectories $\{x_k\}_{k=1}^n$ using $\pi_{\theta^{(j)}}$ and compute sample (sub)gradients:

$$\widehat{\nabla}_\lambda \widehat{\mathcal{L}}(\pi_{\theta^{(j)}}, t^{(j)}, \lambda^{(j)}), \quad \widehat{\nabla}_t \widehat{\mathcal{L}}(\pi_{\theta^{(j)}}, t^{(j)}, \lambda^{(j)}).$$

6:     Update dual ($\lambda$) and auxiliary ($t$) variables:

$$\lambda^{(j+1)} \leftarrow \Pi_\Lambda \left( \lambda^{(j)} - \eta_\lambda \widehat{\nabla}_\lambda \widehat{\mathcal{L}}(\pi_{\theta^{(j)}}, t^{(j)}, \lambda^{(j)}) \right),$$
$$t^{(j+1)} \leftarrow \Pi_\mathcal{T} \left( t^{(j)} + \eta_t \widehat{\nabla}_t \widehat{\mathcal{L}}(\pi_{\theta^{(j)}}, t^{(j)}, \lambda^{(j)}) \right).$$

7: **end for**
8: **Return:** $\pi_{\theta^{(j^*)}}$, where we sample $j^* \sim \mathrm{Unif}([J])$.

---

(10) reduces to an unconstrained RL problem with a linear combination of terms defined by reward functions $r_i'$, given $\lambda$ and $t$. Hence, we have a practical algorithm for solving (10) which, if solved exactly for a starting point $t \in \mathcal{I}$, is arbitrarily close to the true primal solution $P^*$ of (6), under Assumption 3.4. In practice, this problem is non-convex so we may only hope to find a locally optimal solution. Convergence to such a solution is established next.

### 3.3 Algorithm and Analysis

Next, we present the algorithmic procedure for solving problem (10), in Algorithm 1. Consider a policy oracle that precisely solves the problem $\sup_{\theta \in \Theta} \mathcal{L}(\pi_\theta, t, \lambda)$, such that for every $(t, \lambda) \in \mathcal{T} \times \Lambda$, it returns a policy $\pi_{\theta^*(t,\lambda)} \in \arg\max_{\theta \in \Theta} \mathcal{L}(\pi_\theta, t, \lambda)$. Using such an oracle, problem (10) reads

$$D_\theta^* = \sup_{t \in \mathcal{T}} \inf_{\lambda \in \Lambda} \underbrace{\mathcal{L}(\pi_{\theta^*(t,\lambda)}, t, \lambda)}_{:= -f(t,\lambda)} = -\inf_{t \in \mathcal{T}} \sup_{\lambda \in \Lambda} f(t, \lambda). \tag{11}$$

Having access to a solver which returns $\pi_{\theta^*(t,\lambda)}$ exactly is difficult. Instead, we assume the availability of an inexact oracle, as defined next. We justify the generality of this assumption in Appendix A.5 to save space in the main body.

**Assumption 3.6** (Local Solutions). *For any $(t, \lambda) \in \mathcal{T} \times \Lambda$, let $\theta^\star(t, \lambda)$ be a maximum of $\mathcal{L}(\pi_\theta, t, \lambda)$ and $\theta^\dagger(t, \lambda)$ be returned by a generic RL algorithm. Then, there exists a $\delta > 0$ such that $\forall(t, \lambda) \in \mathcal{T} \times \Lambda$, $\widehat{\nabla}_{t,\lambda} \widehat{\mathcal{L}}(\pi_{\theta^\dagger(t,\lambda)}, t, \lambda) = \widehat{\nabla}_{t,\lambda} \widehat{\mathcal{L}}(\pi_{\theta^*(t,\lambda)}, t, \lambda) + b(\theta^*, \theta^\dagger, t, \lambda)$, with $\|b(\theta^*, \theta^\dagger, t, \lambda)\| \leq \delta$.*

In what follows, we show that (10) can be reduced to a stochastic minimax optimization problem satisfying appropriate assumptions, and thus we can obtain a non-asymptotic convergence rate for Algorithm 1. For notational convenience, we define the function $\Phi(t) := \sup_{\lambda \in \Lambda} f(t, \lambda)$. First, we show that $f$ is Lipschitz with respect to $t \in \mathbb{R}^{m+1}$ uniformly over $\lambda \in \Lambda$.

**Lemma 3.7.** *There exists a constant $C > 0$ such that*

$$\left| \sup_\pi \mathcal{L}(\pi, t_1, \lambda) - \sup_\pi \mathcal{L}(\pi, t_2, \lambda) \right| \leq C \|t_1 - t_2\|_2$$

*for all pairs $t_1, t_2 \in \mathbb{R}^{m+1}$ and $\lambda \in \Lambda$.*

We prove Lemma 3.7 (which readily applies to parametrized policies as well) in Appendix A.6. Following Lin et al. [25], we analyze our method under the assumption of Lipschitz smoothness of the function $\mathcal{L}(\pi_{\theta^*(t,\lambda)}, t, \lambda)$, for $\theta^*(t, \lambda) \in \arg\max_{\theta \in \Theta} \mathcal{L}(\pi_\theta, t, \lambda)$, which we state below.

**Assumption 3.8** (Lipschitz Smoothness). *The function $f(t, \lambda) \equiv -\mathcal{L}(\pi_{\theta^*(t,\lambda)}, t, \lambda)$ is $\ell$-smooth over $\mathcal{T} \times \Lambda$, where $\theta^*(t, \lambda) \in \arg\max_{\theta \in \Theta} \mathcal{L}(\pi_\theta, t, \lambda)$ is an arbitrary selection.*

*Remark* 3.9. We note that Assumption 3.8 is not particularly restrictive in our setting, and it holds under several general conditions, as outlined in Appendix A.7. Nonetheless, if not satisfied, we can still enforce this assumption by making a slight algorithmic adjustment involving the appropriate addition of small uniform noise. For a comprehensive discussion we again refer the reader to Appendix A.7, where we also present an explicit formula for computing the (exact) sample gradients.

As appears in prior work [25], we make the standard (and, in our case, mild) assumption that the exact stochastic (sub)gradient oracles are unbiased and have at most $\sigma^2$ variance.

**Assumption 3.10** ((Sub)gradient oracles)**.** For all $(t, \lambda) \in \mathcal{T} \times \Lambda$, we assume that

$$\mathbb{E}[\widehat{\nabla}_{t,\lambda}\widehat{\mathcal{L}}(\pi^*(t,\lambda),t,\lambda) + \nabla f(\lambda,t)] = 0, \quad \mathbb{E}[\|\widehat{\nabla}_{t,\lambda}\widehat{\mathcal{L}}(\pi^*(t,\lambda),t,\lambda) + \nabla f(\lambda,t)\|_2^2] \leq \sigma^2.$$

Given the imposed assumptions, we now present the convergence guarantees of Algorithm 1. In the context of nonsmooth weakly convex optimization, the Moreau envelope serves as a measure of closeness to stationarity. With this in mind, we introduce the definition of an $\epsilon$-stationary point.

**Definition 3.11** ($\epsilon$-stationary point)**.** A point $x$ is $\epsilon$-stationary if $\|\nabla\Phi_{1/2\ell}(x)\| \leq \epsilon$, where, given some $\delta > 0$, we define the Moreau envelope as $\Phi_\delta(x) := \inf_{w \in \mathbb{R}^{m+1}} \left\{ \Phi(w) + \frac{1}{2\delta}\|w - x\|^2 \right\}$.

When $\epsilon = 0$, a stationary point of the Moreau envelope $\Phi_\delta(\cdot)$ is also stationary for $\Phi(\cdot)$. For $\epsilon > 0$, one can argue that an $\epsilon$-stationary point for $\Phi_{1/2\ell}$ is close to an $\epsilon$-stationary point of $\Phi(\cdot)$, as long as $\Phi(\cdot)$ is $\ell$-weakly convex (noting that, under our assumptions, $\Phi(\cdot)$ is $\ell$-weakly convex). We proceed with a bound on the iteration complexity of Algorithm 1.

**Theorem 3.12.** *Let the step-sizes $\eta_t$ and $\eta_\lambda$ in Algorithm 1 be small enough as in Eq. (30) and (29), respectively, and batch size $n = 1$. Define the quantities $\widehat{\Delta}_\Phi := \Phi_{1/2\ell}(t^{(0)}) - \min_t \Phi_{1/2\ell}(t)$ and $\widehat{\Delta}_0 := \Phi(t^{(0)}) - f(\lambda^{(0)}, t^{(0)})$. If Assumptions 3.6, 3.8 and 3.10 hold, the iteration complexity for recovering an $\mathcal{O}(\sqrt{\epsilon^2 + \delta\ell(\mathrm{diam}(\mathcal{T}) + \mathrm{diam}(\Lambda))})$-stationary point (Definition 3.11) is of order*

$$\mathcal{O}\left(\left(\frac{\ell^3(C^2 + \sigma^2 + \delta^2)(\mathrm{diam}(\Lambda))^2 \cdot \widehat{\Delta}_\Phi}{\epsilon^6} + \frac{\ell^3(\mathrm{diam}(\Lambda))^2 \cdot \widehat{\Delta}_0}{\epsilon^4}\right) \max\left\{1, \frac{\sigma^2 + \delta^2}{\epsilon^2}\right\}\right).$$

Note that this result says we only need single trajectories ($n = 1$). Thus our algorithm can be run *online*. The proof, which is detailed in Appendix A.8, extends the developments of [25], by incorporating biased (sub)gradient samples, occurring due to the use of inexact oracles. If $\delta = \mathcal{O}(\epsilon^2)$, then we recover an $\epsilon$-stationary point and the complexity of [25].

# 4 Experiments

We conduct experiments on locomotion tasks from the Safety-Gymnasium benchmark [19].

## 4.1 Setup

We consider two constraints in the environment: safe navigation and safe velocity. The first involves simple problems with low-dimensional state and action spaces, where the agent must navigate under rules (e.g., along a path or without hitting obstacles). The second is more challenging, involving standard high-dimensional control tasks based on MuJoCo-v4 agents [46]. At each timestep, the agent receives a cost of +1 if it violates a constraint and 0 otherwise. These violations do not affect environment dynamics—they neither terminate episodes nor alter transitions. Importantly, our experiments do not access this cost directly during training; instead, the agent manages risk by being risk-aware in its velocities.

To simulate risk and uncertainty in an otherwise deterministic setting (where identical actions yield identical outcomes), we inject zero-mean Gaussian noise with a standard deviation of 0.05 (5% of the action range) into all agent actions during both training and evaluation. This adds stochasticity to the agent's actions and introduces risk.

**Experimental Goals and Motivation** Our objective is to evaluate whether the proposed risk-sensitive method ensures stable convergence of the dual variable ($\lambda$) and the auxiliary variable ($t$), while also inducing safe and meaningful behavioral changes to the agent. To emphasize convergence, we train agents for extended durations (5–18M steps) to provide a thorough *proof-of-work*. Specifically, we aim to: (1) verify that $t$ converges to the empirical $\mathrm{CVaR}^\beta$ of the post-training constraint violation distribution (e.g., safe velocity violations), (2) confirm that $\lambda$ stabilizes to enforce the constraint, and (3) assess whether the policy maximizes reward while maintaining safe velocities.

Table 2: Cumulative episodic evaluation costs and rewards of the converged agents, computed as the mean of the last 10 episodes. The PPO baseline is unconstrained and risk-neutral. We use the simplest Safety-Gymnasium agent, Point at level 1 difficulty, to avoid additional challenges and isolate the effects of the constraints.

| Environment | # Training Steps | Cost ↓ | | Reward ↑ | |
|---|---|---|---|---|---|
| | | PPO | Ours | PPO | Ours |
| Button | 5M | 150.76 | 0.0 | 24.29 | 2.58 |
| Circle | 5M | 206.74 | 0.0 | 60.18 | 39.19 |
| Goal | 10M | 45.09 | 0.0 | 21.89 | 13.56 |
| Push | 5M | 38.48 | 0.0 | 0.93 | 2.42 |

**Optimization with CVaR Constraints**   Let $\upsilon : \mathcal{S} \times \mathcal{A} \to \mathbb{R}$ denote the constraint-quantifying function (e.g., velocity). We formulate the following constrained optimization problem:

$$\sup_{\pi \in \mathcal{P}(\mathcal{S})} \mathbb{E} \left[ \sum_{\tau=0}^{\infty} \gamma^{\tau} r(s_{\tau}, \pi(s_{\tau})) \right] \quad \text{s.t.} \quad \text{CVaR}_{\nu\pi}^{\beta}(\upsilon(s, a)) \leq c,$$

where $c$ is a user-defined threshold and $\beta$ is the CVaR parameter. Our framework is general and allows risk-neutral objectives to be combined with risk-aware constraints. To solve this, we transform the reward at each timestep using the CVaR-based Lagrangian (see Appendix B.3.1 for the derivation):

$$r(s_{\tau}, \pi(s_{\tau})) + \lambda_i \left( c + t - \frac{1}{\beta} \left( t + \upsilon(s_{\tau}, \pi(s_{\tau})) \right)_+ \right).$$

In safe navigation, the constraint variable reduces to $\upsilon(s, a) \in \{0, 1\}$, since navigation is not quantifiable in the same way as velocity. This makes the optimization problem more difficult, as the penalty term (the $\lambda$ multiplier) becomes sparse.

Proximal Policy Optimization (PPO) [39] is used as a black-box solver to optimize the policy $\pi$, implementing line 4 of Algorithm 1. The dual variable $\lambda$ and auxiliary $t$ are updated via stochastic gradient steps following the rest of Algorithm 1. For further experimental details, see Appendix B.

### 4.2   Results

Results are shown in Table 2 for safe navigation and in Figure 1 for safe velocity.

**Constraint Handling**   The performance in the navigation tasks shows us that our method can strictly prevent any constraint violations, whether quantified continuously or not. In fact, it is the only method achieving strictly zero violations compared to other PPO-based constraint learning algorithms (see Table 5 in [19]). In some cases, constraint satisfaction also leads to higher rewards than the unconstrained vanilla agent (e.g., in the Push environment). This shows that constraints are not arbitrary but can guide the agent toward optimal task behavior.

**Risk-Management**   On the other hand, the velocity tasks provide insights into the risk management capabilities. With properly tuned step sizes, the dual and CVaR variables stabilize and oscillate around consistent values. The post-training evaluations (the last column in Figure 1) extract the distribution of the trained agent's velocity over several evaluation episodes. We observe that the converged $t$-value matches the $\beta$-upper quantile of the velocity distribution. This is what the CVaR aims to do: it captures the expected cost in the worst $\beta$ fraction of outcomes.

Further, the evaluation reward becomes increasingly stable as training progresses despite the uncertainty in the evaluation environment. Considering the number of data points in the curves—1000 per 1M steps (e.g., about 14,000 points in Hopper)—the robustness of our method in managing risk is evident. This is also confirmed in post-training simulations: the agent moves forward cautiously compared to the unconstrained (risk-neutral) PPO agent, demonstrating that the learned policy adheres to the task while remaining interpretable from the perspective of the constraint.

We recognize the extended training duration and the need to tune the step sizes of $\lambda$ and $t$. Nonetheless, once these variables converge, the algorithm is effective not only in toy tasks but also in realistic control settings, making patience during training an important practical consideration.

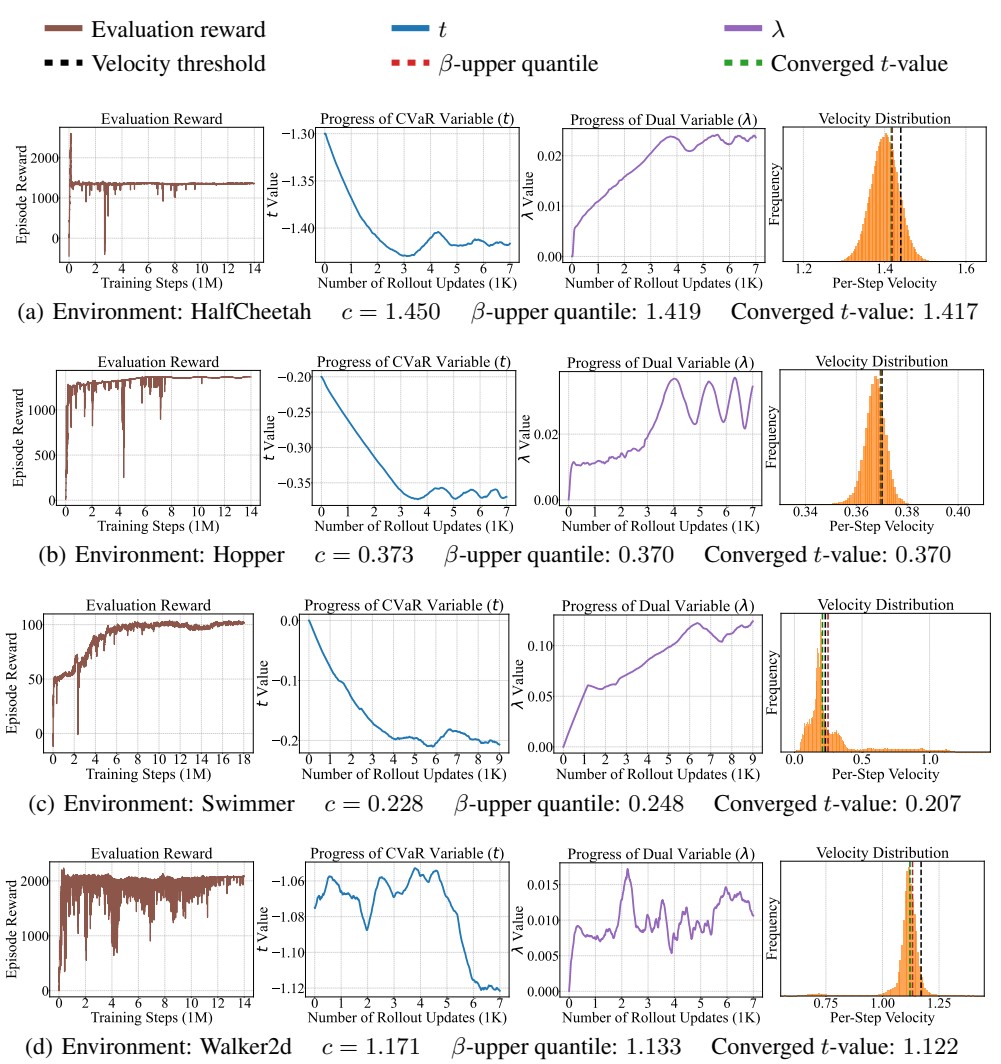

Figure 1: Learning curves of the agent trained to convergence. We report episodic evaluation rewards, the progression of optimization variables $t$ and $\lambda$, and the velocity distribution of the trained agent evaluated over 100 post-training episodes without further learning. Curves are *not* smoothed.

## 5    Conclusion

We introduce a training framework for solving constrained reinforcement learning (RL) problems with risk-awareness which has desirable properties. The problem exhibits a parameterized strong duality under constraint qualifications, which lends itself to an algorithm that is theoretically-supported and flexible to implement; the inner problem can be solved using any black-box RL algorithm, while the other variables are updated using SGDA. The framework handles general OCEs beyond the CVaR, and captures both risk-neutral (e.g., by setting $\beta = 1$ for the $\text{CVaR}^\beta$) and risk-aware objectives and constraints, or a combination of the two, with empirical evidence that, in the context of risk-aware constrained RL, it effectively manages risk.

**Limitations**    An open problem resulting from this work asks whether or not full strong duality holds unconditionally. Also, while our framework is inherently risk-aware, it is more computationally intensive compared to risk-neutral methods (as each update of $\lambda$ and $t$ variables requires solving for an approximately optimal policy), as is also true for other risk-aware methods, e.g., [7, 12].

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

# A Supporting Details for Theoretical Results

## A.1 Robustness in Value and Time

The so-called *reward-based* formulation in (4), as coined by Bisi et al. [6], Bonetti et al. [7], captures *per-stage risk*, in contrast to the return-based approach (3), which captures aggregate discounted risk. To see this, first note that the occupation measure can be reexpressed as

$$\mathrm{d}\nu^\pi = \sum_{\tau=0}^\infty (1-\gamma)\gamma^\tau \mathrm{d}p_\pi^\tau =: \sum_{\tau=0}^\infty \mathrm{d}p_\pi^{T=\tau} p^T(\tau),$$

where $T$ is a $\mathbb{N} \cup \{0\}$-valued geometric random variable such that $P(T = \tau) =: p^T(\tau) := (1-\gamma)\gamma^\tau, \tau \in \mathbb{N} \cup \{0\}$, and $p_\pi^T$ is now the conditional state-action distribution relative to $T$. In other words, $\nu^\pi$ may be seen as the marginalization of the disintegration $\mathrm{d}p_\pi^{T=\tau}(s,a)p^T(\tau)$ relative to $T$, the latter interpreted as a random time. One can show that

$$-\mathrm{CVaR}_{\nu^\pi}^\beta(-r(s,a)) = -\mathrm{CVaR}^\beta(-r(s_T^\pi, a_T^\pi)),$$

where $(s_T^\pi, a_T^\pi)$ denotes the state-action vector under policy $\pi$ and evaluated at random time $T$. Specifically for the case of CVaR (and similarly for other OCEs), it can be readily seen that for $\beta = 1$ we recover the objective of (1) which can be equivalently written as $V(\pi) = (1-\gamma)^{-1}\mathbb{E}\{r(s_T^\pi, a_T^\pi)\}$, whereas if $\beta \downarrow 0$, we obtain

$$\lim_{\beta\downarrow 0} -\mathrm{CVaR}_{\nu^\pi}^\beta(-r(s,a)) = \inf_\tau \operatorname{ess\,inf}_{(s,a)\sim p_\pi^\tau} r(s,a),$$

which shows in particular that such reward-based RL formulations enforce *joint aversion in reward value and the time* for which the reward takes each value. For instance, for sufficiently small $\beta$, the functional $-\mathrm{CVaR}^\beta(-r(s_T^\pi, a_T^\pi))$ will be more sensitive to very small rewards happening far in the future (and with low probability $(1-\gamma)\gamma^\tau$), as compared with (1) and (3), which both consider reward time averaging, heavily discounting future reward contributions.

## A.2 Proof of Lemma 3.2

We will work with the convex analytic dual form for this proof, but replacing the occupancy measure with the expected discount sum will give the same result.

**Statement** The problem

$$P^* = \sup_{\pi\in\mathcal{R}} \sup_{t_0\in\mathbb{R}} \mathbb{E}_{\nu^\pi(s,a)}\left[t_0 - \frac{1}{\beta}(t_0 - r_0(s,a))_+\right]$$

$$\text{s.t.} \sup_{t_i\in\mathbb{R}} \mathbb{E}_{\nu^\pi(s,a)}\left[t_i - \frac{1}{\beta}(t_i - r_i(s,a))_+\right] \geq c_i \qquad \forall i = 1, 2, \ldots, m \tag{12}$$

and that of

$$\sup_{\pi\in\mathcal{R},t_0\in\mathbb{R},t_i\in\mathbb{R}} \mathbb{E}_{\nu^\pi(s,a)}\left[t_0 - \frac{1}{\beta}(t_0 - r_0(s,a))_+\right]$$

$$\text{s.t.} \quad \mathbb{E}_{\nu^\pi(s,a)}\left[t_i - \frac{1}{\beta}(t_i - r_i(s,a))_+\right] \geq c_i \qquad \forall i = 1, 2, \ldots, m \tag{13}$$

are equivalent.

*Proof.* We need to prove that it is valid to move the supremum over each $t_i$ to the objective. The proof follows that of Chow et al. [12].

- (Case (6) $\leq$ (7).) Let $\pi^1$ be a feasible policy for the problem (6). Then define

$$t_i^1 := \arg\max_{t_i\in\mathbb{R}} \mathbb{E}_{\nu^{\pi^1}(s,a)}\left[t_i - \frac{1}{\beta}(t_i - r_i(s,a))_+\right] \geq c_i, \qquad \forall i = 1, 2, \ldots, m,$$

  so that $(\pi^1, t_i^1)$ is feasible for (6). Then this tuple $(\pi^1, t_i^1)$ is clearly also feasible for (7).

- (Case (7) ≤ (6).) On the other hand, for any $\pi$, we clearly have that

$$-\text{CVaR}^{\beta}_{\nu^{\pi}}[-Z] = \sup_{t} \mathbb{E}_{\nu^{\pi}}\left[t - \frac{1}{\beta}(t - Z)_{+}\right] \geq \mathbb{E}_{\nu^{\pi}}\left[\widehat{t} - \frac{1}{\beta}(\widehat{t} - Z)_{+}\right],$$

for any $\widehat{t}$. So, if the quantity on the RHS of the inequality is lower bounded by $c_i$ (that is, feasible for (7)), then the LHS is clearly also lower bounded by $c_i$ (thus, feasible for (6)).

$\square$

## A.3 Proof of Proposition 3.3

**Statement** Let $r_i$ be bounded functions for all $i \in \{0\} \cup [m]$. Assume that Slater's condition holds for (9). Then, (9) exhibits strong duality, and thus

$$\sup_{\pi} \inf_{\lambda \geq 0} \mathcal{L}(\pi, t, \lambda) = \inf_{\lambda \geq 0} \sup_{\pi} \mathcal{L}(\pi, t, \lambda).$$

*Proof.* Paternain et al. [33, Theorem 1] proved strong duality holds for problems of the form

$$\sup_{\pi \in \mathcal{P}(\mathcal{S})} \mathbb{E}\left[\sum_{\tau=0}^{\infty} \gamma^{\tau} r_0(s_{\tau}^{\pi}, a_{\tau}^{\pi})\right] \text{ s.t. } \mathbb{E}\left[\sum_{\tau=0}^{\infty} \gamma^{\tau} r_i(s_{\tau}^{\pi}, a_{\tau}^{\pi})\right] \geq c_i, \ \forall i \in [m], \tag{14}$$

which involve the expectation of the discounted sums of some bounded reward functions $r_i$, for $i \in \{0\} \cup [m]$. Thus, the problem (9) is exactly in the form of Paternain et al. [33], with the reward functions $r_i'(s, a, t_i) = t_i - \frac{1}{\beta}(t_i - r_i(s, a))_+$, for $i \in \{0\} \cup [m]$. Since the original $r_i$ are bounded and $t_i$ are fixed, we have that $r_i'$ are bounded and we can readily apply Paternain et al. [33, Theorem 1]. $\square$

## A.4 Proof of Theorem 3.5

Under Assumption 3.4, we have shown that the proposed partial Lagrangian relaxation is exact, i.e., that

$$\sup_{t \in \mathcal{I}} \sup_{\pi} \inf_{\lambda \geq 0} \mathcal{L}(\pi, t, \lambda) = \sup_{t \in \mathcal{I}} \inf_{\lambda \geq 0} \sup_{\pi} \mathcal{L}(\pi, t, \lambda).$$

We assume that $\pi_{\theta}$, for $\theta \in \Theta \subset \mathbb{R}^p$ (with $\Theta$ some compact set), is an $\epsilon$-universal parametrization of measures in $\mathcal{P}(\mathcal{S})$, according to Paternain et al. [33, Definition 1] and that the parametrized-policy version of (7) is feasible. We proceed to show that there is almost no price to pay for the policy parametrization, in terms of duality gap.

Indeed, assuming that the parametrized optimization problem is feasible, we can utilize Paternain et al. [33, Theorem 2] and Assumption 3.4 to deduce that, for all $t \in \mathcal{I}$, the following inequalities hold:

$$P^*(t) := \sup_{\pi} \inf_{\lambda \geq 0} \mathcal{L}(\pi, t, \lambda) \geq \inf_{\lambda \geq 0} \sup_{\theta} \mathcal{L}(\pi_{\theta}, t, \lambda) \geq P^*(t) - G(t)\epsilon/(1 - \gamma),$$

where $G(t) = \mathcal{O}(1)$ is a constant depending on $t$, and $\epsilon > 0$ is an arbitrarily small constant (depending on the employed parametrization). Let $t^* \in \arg\max_{t \in \mathcal{I}} P^*(t)$ and $t^*_{\theta} \in \arg\max_{t \in \mathcal{I}} \inf_{\lambda \geq 0} \sup_{\theta} \mathcal{L}(\pi_{\theta}, t, \lambda)$. Let also $G = \max\{G(t^*), G(t^*_{\theta})\}$. Then, we obtain that

$$P^*(t^*) \geq \inf_{\lambda \geq 0} \sup_{\pi} \mathcal{L}(\pi, t^*, \lambda) \geq P^*(t^*) - G\epsilon/(1 - \gamma),$$

$$P^*(t^*_{\theta}) \geq \inf_{\lambda \geq 0} \sup_{\theta} \mathcal{L}(\pi_{\theta}, t^*_{\theta}, \lambda) \geq P^*(t^*_{\theta}) - G\epsilon/(1 - \gamma).$$

Since, by definition, $P^*(t^*) \geq P^*(t^*_{\theta})$, while $\sup_{\theta} \inf_{\lambda \geq 0} \mathcal{L}(\pi_{\theta}, t^*, \lambda) \leq \sup_{\theta} \inf_{\lambda \geq 0} \mathcal{L}(\pi_{\theta}, t^*_{\theta}, \lambda)$, we obtain that

$$P^*(t^*) \equiv \sup_{t \in \mathcal{I}} \sup_{\pi} \inf_{\lambda \geq 0} \mathcal{L}(\pi, t, \lambda) \geq \sup_{t \in \mathcal{I}} \inf_{\lambda \geq 0} \sup_{\theta} \mathcal{L}(\pi_{\theta}, t, \lambda) \geq P^*(t^*) - \mathcal{O}\left(\frac{\epsilon}{1 - \gamma}\right),$$

which completes the proof. $\square$

## A.5 Discussion on Assumption 3.6

Below, we provide an insight as to why the inexact oracle utilized in Assumption 3.6 is reasonable under minimal conditions (under which the sample gradients used in Algorithm 1 satisfy Assumption 3.6). Specifically, building upon the discussion given later in Appendix A.7, we note that the most general condition guaranteeing Assumption 3.8 is given in the second scenario of Appendix A.7, which requires that the function $\mathcal{L}(\pi_{\theta^*}(t,\lambda), t, \lambda)$ satisfies the Lojasiewicz inequality with uniform constants over $(t, \lambda)$, for any selection $\pi_{\theta^*}(t, \lambda) \in \arg\max_{\theta \in \Theta} \mathcal{L}(\pi_\theta, t, \lambda)$ (again, see Appedinx A.7 for the definition of the Lojasiewicz inequality in this context). Since the (parametrized) policy optimization problem $\max_{\theta \in \Theta} \mathcal{L}(\pi_\theta, t, \lambda)$ is nonconvex (and is approximately solved using a solver like PPO), we can only expect to obtain an approximate solution $\theta^\dagger(t, \lambda)$ such that

$$\mathcal{L}(\pi_{\theta^*}(t,\lambda), t, \lambda) - \mathcal{L}(\pi_{\theta^\dagger(t,\lambda)}, t, \lambda) \le \widetilde{\varepsilon}(\theta^\dagger, \theta^*, t, \lambda) \le \varepsilon,$$

for some $\varepsilon > 0$ (uniformly bounded in $(t, \lambda)$). Then, if $\mathcal{L}(\pi_{\theta^*}(t,\lambda), t, \lambda)$ satisfies the Lojasiewicz inequality for some uniform positive constants $C, \eta$, we obtain that

$$\|\theta^\dagger - \theta^*\| \le C\varepsilon^\eta.$$

Thus, under the mere assumption that the sample gradient function $\widehat{\nabla}_{t,\lambda}\widehat{\mathcal{L}}(\pi_\theta, t, \lambda)$ is $\widehat{L}$-Lipschitz continuous with respect to $\theta$ (which can be enforced, if necessary, by appropriate smoothing of the utility function $g$; a general smoothing strategy for generating smooth OCEs from non-smooth utilities can be found in Kouri and Surowiec [22, Example 2]), we obtain that

$$\left\|\widehat{\nabla}_{t,\lambda}\widehat{\mathcal{L}}(\pi_{\theta^\dagger(t,\lambda)}, t, \lambda) - \widehat{\nabla}_{t,\lambda}\widehat{\mathcal{L}}(\pi_{\theta^*(t,\lambda)}, t, \lambda)\right\| \le \widehat{L}\|\theta^\dagger - \theta^*\| \le \widehat{L}C\varepsilon^\eta := \delta.$$

Upon noting that $\widehat{\nabla}_{t,\lambda}\widehat{\mathcal{L}}(\pi_{\theta^*(t,\lambda)}, t, \lambda)$ is an unbiased sample of $-\nabla_{t,\lambda}f(t, \lambda)$ (cf. Appendix A.7), we can easily deduce that under the said minor conditions, the sample gradients utilized in Algorithm 1 satisfy Assumption 3.6. Thus, we can see that our oracle condition in Assumption 3.6 is justified and can be shown to hold under minimal conditions (without requiring convexity or uniqueness of the solution of $\max_{\theta \in \Theta} \mathcal{L}(\pi_\theta, t, \lambda)$).

## A.6 Lipschitzness of the Lagrangian

First we show a technical lemma, and then we proceed with the proof of Lemma 3.7.

**Lemma A.1.** *For any pair $(\widetilde{t}, \overline{t}) \in \mathbb{R}^{2m+2}$, and for any $\lambda \in \mathbb{R}^m$, it is true that*

$$|\sup_\pi \mathcal{L}(\pi, \widetilde{t}, \lambda) - \sup_\pi \mathcal{L}(\pi, \overline{t}, \lambda)| \le \sup_\pi |\mathcal{L}(\pi, \widetilde{t}, \lambda) - \mathcal{L}(\pi, \overline{t}, \lambda)| \tag{15}$$

**Proof of Lemma A.1**   For any pair $\widetilde{t}, \overline{t} \in \mathbb{R}$ it is true that

$$\sup_\pi \mathcal{L}(\pi, \widetilde{t}, \lambda) = \sup_\pi \mathcal{L}(\pi, \widetilde{t}, \lambda) - L(\pi, \overline{t}, \lambda) + L(\pi, \overline{t}, \lambda) \tag{16}$$

$$\le \sup_\pi \mathcal{L}(\pi, \widetilde{t}, \lambda) - \mathcal{L}(\pi, \overline{t}, \lambda) + \sup_\pi \mathcal{L}(\pi, \overline{t}, \lambda) \implies \tag{17}$$

$$\sup_\pi \mathcal{L}(\pi, \widetilde{t}, \lambda) - \sup_\pi \mathcal{L}(\pi, \overline{t}, \lambda) \le \sup_\pi \mathcal{L}(\pi, \widetilde{t}, \lambda) - \mathcal{L}(\pi, \overline{t}, \lambda) \le \sup_\pi |\mathcal{L}(\pi, \widetilde{t}, \lambda) - \mathcal{L}(\pi, \overline{t}, \lambda)| \tag{18}$$

Similarly, we can show that

$$\sup_\pi \mathcal{L}(\pi, \overline{t}, \lambda) - \sup_\pi \mathcal{L}(\pi, \widetilde{t}, \lambda) \le \sup_\pi \mathcal{L}(\pi, \overline{t}, \lambda) - \mathcal{L}(\pi, \widetilde{t}, \lambda) \le \sup_\pi |\mathcal{L}(\pi, \overline{t}, \lambda) - \mathcal{L}(\pi, \widetilde{t}, \lambda)|. \tag{19}$$

The last two displays give the inequality (15).

**Proof of Lemma 3.7**   For any pair $(\widetilde{t}, \overline{t}) \in \mathbb{R}^{2m+2}$, we find an upper bound on the term $|\mathcal{L}(\pi, \widetilde{t}, \lambda) - \mathcal{L}(\pi, \overline{t}, \lambda)|$ as follows

$$|\mathcal{L}(\pi, \widetilde{t}, \lambda) - \mathcal{L}(\pi, \overline{t}, \lambda)|$$

$$\le \mathbb{E}\left[\sum_{\tau=0}^\infty \gamma^\tau \left|\left(\widetilde{t}_0 - \frac{1}{\beta}(\widetilde{t}_0 - r_0(s_\tau, a_\tau))_+\right) - \left(\overline{t}_0 - \frac{1}{\beta}(\overline{t}_0 - r_0(s_\tau, a_\tau))_+\right)\right|\right]$$

$$+ \mathbb{E}\left[\sum_{\tau=0}^{\infty}\gamma^\tau\left|\sum_{i=1}^m \lambda_i\left(\widetilde{t}_i - \frac{1}{\beta_i}(\widetilde{t}_i - r_i(s_\tau,a_\tau))_+\right)\right.\right.$$
$$\left.\left. - \sum_{i=1}^m \lambda_i\left(\overline{t}_i - \frac{1}{\beta_i}(\overline{t}_i - r_i(s_\tau,a_\tau))_+\right)\right|\right]$$

$$= \mathbb{E}\left[\sum_{\tau=0}^{\infty}\gamma^\tau\left|\left(\widetilde{t}_0 - \frac{1}{\beta}\max\{\widetilde{t}_0 - r_0(s_\tau,a_\tau),0\}\right) - \left(\overline{t}_0 - \frac{1}{\beta}\max\{\overline{t}_0 - r_0(s_\tau,a_\tau),0\}\right)\right|\right]$$

$$+ \mathbb{E}\left[\sum_{\tau=0}^{\infty}\gamma^\tau\left|\sum_{i=1}^m \lambda_i\left(\widetilde{t}_i - \frac{1}{\beta_i}\max\{\widetilde{t}_i - r_i(s_\tau,a_\tau)\}\right)\right.\right.$$
$$\left.\left. - \sum_{i=1}^m \lambda_i\left(\overline{t}_i - \frac{1}{\beta_i}\max\{\overline{t}_i - r_i(s_\tau,a_\tau),0\}\right)\right|\right]$$

$$\leq \mathbb{E}\left[\sum_{\tau=0}^{\infty}\gamma^\tau(|\widetilde{t}_0 - \overline{t}_0| + \frac{1}{\beta}|\max\{\widetilde{t}_0 - r_0(s_\tau,a_\tau),0\} - \max\{\overline{t}_0 - r_0(s_\tau,a_\tau),0\}|)\right]$$

$$+ \mathbb{E}\left[\sum_{\tau=0}^{\infty}\gamma^\tau\sum_{i=1}^m\lambda_i(|\widetilde{t}_i - \overline{t}_i| + \frac{1}{\beta_i}|\max\{\widetilde{t}_i - r_i(s_\tau,a_\tau),0\} - \max\{\overline{t}_i - r_i(s_\tau,a_\tau),0\}|)\right]$$

$$\leq \mathbb{E}\left[\sum_{\tau=0}^{\infty}\gamma^\tau(|\widetilde{t}_0 - \overline{t}_0| + \frac{1}{\beta}\max\{|\widetilde{t}_0 - r_0(s_\tau,a_\tau) - \overline{t}_0 - r_0(s_\tau,a_\tau)|,0\})\right]$$

$$+ \mathbb{E}\left[\sum_{\tau=0}^{\infty}\gamma^\tau\sum_{i=1}^m\lambda_i(|\widetilde{t}_i - \overline{t}_i| + \frac{1}{\beta_i}\max\{|\widetilde{t}_i - r_i(s_\tau,a_\tau) - \overline{t}_i - r_i(s_\tau,a_\tau)|,0\})\right]$$

$$= \mathbb{E}\left[\sum_{\tau=0}^{\infty}\gamma^\tau(|\widetilde{t}_0 - \overline{t}_0| + \frac{1}{\beta}|\widetilde{t}_0 - \overline{t}_0|)\right] + \mathbb{E}\left[\sum_{\tau=0}^{\infty}\gamma^\tau\sum_{i=1}^m\lambda_i(|\widetilde{t}_i - \overline{t}_i| + \frac{1}{\beta_i}|\widetilde{t}_i - \overline{t}_i|)\right]$$

$$\leq (1-\gamma)^{-1}(1+\frac{1}{\beta})|\widetilde{t}_0 - \overline{t}_0| + (1-\gamma)^{-1}\sum_{i=1}^m\lambda_i(1+\frac{1}{\beta_i})|\widetilde{t}_i - \overline{t}_i|$$

$$\leq (1-\gamma)^{-1}\|\widetilde{t} - \overline{t}\|\sqrt{(1+\frac{1}{\beta})^2 + \sum_{i=1}^m\lambda_i^2(1+\frac{1}{\beta_i})^2}.$$

Then, we apply Lemma A.1 to get

$$|\sup_\pi \mathcal{L}(\pi,\widetilde{t},\lambda) - \sup_\pi \mathcal{L}(\pi,\overline{t},\lambda)| \leq \sup_\pi |\mathcal{L}(\pi,\widetilde{t},\lambda) - \mathcal{L}(\pi,\overline{t},\lambda)| \tag{20}$$

$$\leq (1-\gamma)^{-1}\|\widetilde{t} - \overline{t}\|\sqrt{(1+\frac{1}{\beta})^2 + \sum_{i=1}^m\lambda_i^2(1+\frac{1}{\beta_i})^2}. \tag{21}$$

Thus the function $\mathcal{L}(\pi,\cdot,\lambda)$ is Lipschitz, with Lipschitz constant

$$c(\gamma,\beta,\lambda) := (1-\gamma)^{-1}\sqrt{(1+\frac{1}{\beta})^2 + \sum_{i=1}^m\lambda_i^2(1+\frac{1}{\beta_i})^2}. \tag{22}$$

The result then follows since we are interested in multipliers $\lambda \in \Lambda$, with $\Lambda$ a convex and compact set (and thus, there exists a constant $C \geq c(\gamma,\beta,\lambda)$, independent of $\lambda$ and $t$, for which $\mathcal{L}(\pi,\cdot,\lambda)$ is Lipschitz continuous for all $(t,\lambda) \in \mathcal{T} \times \Lambda$). $\qquad\square$

### A.7 Conditions That Guarantee Assumption 3.8

Let us note that Assumption 3.8 is not particularly restrictive in our setting. In what follows, we discuss some general conditions under which this holds readily, as well as a simple methodology of

enforcing this assumption, if necessary, by a slight algorithmic adjustment. Before we do this, let us first note that $\mathcal{L}(\pi_\theta, \cdot, \lambda)$ is not necessarily differentiable (unless the probability of $t_i - r_i(s, a) = 0$ is 0), because of the plus function $(\cdot)_+$. Nonetheless, adding a small random noise to the reward following a random variable with a smooth mollifier density ensures that the function is infinitely differentiable with respect to both $t$ and $\lambda$. Note that, since we assume that the reward is bounded, we cannot (in theory) add Gaussian noise, but any smooth and compactly supported mollifier-based noise is admissible. Thus, we assume (without loss of generality) that $\mathcal{L}(\pi_\theta, \cdot, \lambda)$ is differentiable (in fact, to an arbitrary degree).

Next, we focus on the task of assessing the differentiability properties of $f(t, \lambda)$. Specifically, we identify three general scenarios that guarantee Assumption 3.8 in our case.

1. The first scenario, which is somewhat restrictive, relies on the assumption that, for each $(t, \lambda) \in \mathcal{T} \times \Lambda$, the maximization problem, with respect to $\theta$, admits a unique solution. In that case, we can readily utilize Lemma 2.2 of Shapiro [40], which states that $\mathcal{L}(\pi_{\theta^*(t,\lambda)}, t, \lambda)$ is twice-continuously differentiable. Since $\mathcal{T} \times \Lambda$ is a compact set, this immediately implies that the function is $\ell$-smooth over $\mathcal{T} \times \Lambda$, for some constant $\ell > 0$.

2. The second scenario is significantly more general, and relies on the strong second-order sufficient optimality conditions for the maximization over $\theta$ and the assumption that the function $\mathcal{L}(\pi_{\theta^*(t,\lambda)}, t, \lambda)$ satisfies the Lojasiewicz inequality with some constant uniform over $(t, \lambda)$, for any $\pi_{\theta^*(t,\lambda)} \in \arg\max_{\theta \in \Theta} \mathcal{L}(\pi_\theta, t, \lambda)$, that is, for each $(t, \lambda) \in \mathcal{T} \times \Lambda$, there exist constants $C > 0$ and $\eta > 0$ such that

$$\text{dist}\left(\theta, \arg\max_{\theta \in \Theta} \mathcal{L}(\pi_\theta, t, \lambda)\right) \leq C \left(\max_{\theta \in \Theta} \mathcal{L}(\pi_\theta, t, \lambda) - \mathcal{L}(\pi_\theta, t, \lambda)\right)^\eta.$$

We note that the previous holds in several cases (typically with uniform exponent $\eta$). For example, this is true when $\max_{\theta \in \Theta} \mathcal{L}(\pi_\theta, \cdot, \cdot)$ is sub-analytic (which, for example, is implied if $\mathcal{L}(\pi_\theta, t, \lambda)$ is analytic, noting that this can easily be enforced our setting). Since $\mathcal{T} \times \Lambda$ is a compact and convex set, we can readily utilize Theorem 11 from [16] to deduce that under this general assumption, the function $\mathcal{L}(\pi_{\theta^*(t,\lambda)}, t, \lambda)$ is $\ell$-smooth over $\mathcal{T} \times \Lambda$, irrespectively of the selections $\theta^*(t, \lambda) \in \arg\max_{\theta \in \Theta} \mathcal{L}(\pi_\theta, t, \lambda)$.

3. Under certain qualification conditions, laid out in Section 4 of Shapiro [40] (that do not require the second-order strong sufficient conditions for the maximization problem over $\theta$), $\ell$-smoothness of $\max_{\theta \in \Theta} \mathcal{L}(\pi_\theta, t, \lambda)$ is also guaranteed. We refer the reader to Shapiro [40] for additional details, since this analysis directly applicable in our setting.

If any of the above conditions holds, then Assumption 3.8 is true, and we can readily compute the gradient of $f$ as

$$\nabla_{t,\lambda} f(t, \lambda) = -\nabla_{t,\lambda} \mathcal{L}(\pi_\theta, t, \lambda)|_{\pi_\theta = \pi_{\theta^*(t,\lambda)}},$$

where $\theta^*(t, \lambda) \in \arg\max_{\theta \in \Theta} \mathcal{L}(\pi_\theta, t, \lambda)$ is an arbitrary selection.

If none of the above is true, we can still guarantee that Assumption 3.8 holds by slightly altering Algorithm 1. Specifically, for some small $\mu > 0$, we can define the surrogate function

$$\mathcal{L}_\mu(\pi_{\theta^*(t,\lambda)}, t, \lambda) \coloneqq \mathbb{E}_{U_1, U_2} \left\{ \mathcal{L}(\pi_{\theta^*(t+\mu U_1, \lambda+\mu U_2)}, t + \mu U_1, \lambda + \mu U_2) \right\}, \tag{23}$$

where $U_1, U_2$ follow a uniform distribution over the unit ball (of appropriate dimension in each case). Then, this new surrogate function can be made arbitrarily close to $\max_{\theta \in \Theta} \mathcal{L}(\pi_\theta, t, \lambda)$ (where the proximity is uniform in the constant $\mu$), while at the same time being $\ell_\mu$-smooth, with $\ell_\mu = \mathcal{O}(1/\mu)$.

Then, we can substitute the original Lagrangian with the smoothed Lagrangian given in (23) and run Algorithm 1 on the surrogate problem. Following standard zeroth-order optimization techniques (e.g., see Pougkakiotis and Kalogerias [34]), we note that, in this case, the only algorithmic adjustments that need to be made relate to the computation of the sample gradients of $f$, which would require two (instead of one) evaluations of the policy oracle. In light of this discussion, we conclude that Assumption 3.8 can be utilized (almost) without loss of generality.

## A.8 Proof of Main Theorem

In this section, we provide the proof of Theorem 3.12. To prove the main theorem we will need the following auxiliary lemmas. The proof extends the analysis given in [25], since the presented algorithm only assumes having access to biased and inexact (sub)gradient samples.

Denote by $b_1(\theta^*, \theta^\dagger, t, \lambda), b_2(\theta^*, \theta^\dagger, t, \lambda)$ the components of $b(\theta^*, \theta^\dagger, t, \lambda)$ corresponding to $t$ and $\lambda$ from Assumption 3.6.

**Lemma A.2.** $\widehat{\nabla}_t \widehat{\mathcal{L}}(\pi_{\theta^\dagger(t,\lambda)}, t, \lambda))$ and $\widehat{\nabla}_\lambda \widehat{\mathcal{L}}(\pi_{\theta^\dagger(t,\lambda)}, t, \lambda))$ have bounded bias and variance.

*Proof.* By Assumption 3.6 and by linearity of expectation, we have

$$
\mathbb{E}\left[\widehat{\nabla}_{t,\lambda}\widehat{\mathcal{L}}(\pi_{\theta^\dagger(t,\lambda)}, t, \lambda))\right] = \mathbb{E}\left[\widehat{\nabla}_{t,\lambda}\widehat{\mathcal{L}}(\pi_{\theta^*(t,\lambda)}, t, \lambda)) + b(\theta^*, \theta^\dagger, t, \lambda)\right]
$$
$$
= -\nabla f(t, \lambda) + \mathbb{E}[b(\theta^*, \theta^\dagger, t, \lambda)]. \qquad \text{(by Assumptions 3.6, 3.10)}
$$

The bias of the gradient is controlled by $b(\theta^*, \theta^\dagger, t, \lambda)$, whose norm is at most $\delta$. Furthermore,

$$
\mathbb{E}\left[\left\|\widehat{\nabla}_{t,\lambda}\widehat{\mathcal{L}}(\pi_{\theta^\dagger(t,\lambda)}, t, \lambda))\right\|^2\right]
$$
$$
= \mathbb{E}\left[\left\|\widehat{\nabla}_{t,\lambda}\widehat{\mathcal{L}}(\pi_{\theta^\dagger(t,\lambda)}, t, \lambda)) - \widehat{\nabla}_{t,\lambda}\widehat{\mathcal{L}}(\pi_{\theta^*(t,\lambda)}, t, \lambda)) + \widehat{\nabla}_{t,\lambda}\widehat{\mathcal{L}}(\pi_{\theta^*(t,\lambda)}, t, \lambda))\right\|^2\right]
$$
$$
\leq 2\mathbb{E}\left[\left\|b(\theta^*, \theta^\dagger, t, \lambda)\right\|^2\right] + 2\mathbb{E}\left[\left\|\widehat{\nabla}_{t,\lambda}\widehat{\mathcal{L}}(\pi_{\theta^*(t,\lambda)}, t, \lambda))\right\|^2\right]
$$
$$
\leq 2\left\|\nabla f(t, \lambda)\right\|^2 + 2(\sigma^2 + \delta^2).
$$
$$
\text{(by Lemma A.2 of [25] with } M = 1 \text{ and Assumptions 3.6, 3.10)}
$$

$\square$

**Lemma A.3.** Let $\Delta^{(k)} = \mathbb{E}\left[\Phi(t^{(k)}) - f(t^{(k)}, \lambda^{(k)})\right]$. The following holds for Algorithm 1:

$$
\mathbb{E}\left[\Phi_{1/2\ell}(t^{(k)})\right] \leq \mathbb{E}\left[\Phi_{1/2\ell}(t^{(k-1)})\right] + 2\eta_t \ell \Delta^{(k-1)} - \frac{\eta_t}{4}\mathbb{E}\left\|\nabla\Phi_{1/2\ell}(t^{(k-1)})\right\|^2
$$
$$
+ 2\eta_t \delta\ell \cdot \operatorname{diam}(\mathcal{T}) + 3\eta_t^2 \ell \cdot (C^2 + \sigma^2 + \delta^2).
$$

*Proof.* Let $\widehat{t}^{(k-1)} = \operatorname{prox}_{\Phi/2\ell}(t^{(k-1)})$ and fix $\pi_{\theta^*} = \pi_{\theta^*(t^{(k-1)}, \lambda^{(k-1)})}$ where $\theta^* \in \arg\max_{\theta \in \Theta} \mathcal{L}(\pi_\theta, t^{(k-1)}, \lambda^{(k-1)})$, and $\pi_{\theta^\dagger} = \pi_{\theta^\dagger(t^{(k-1)}, \lambda^{(k-1)})}$ satisfying Assumption 3.6. Then,

$$
\left\|\widehat{t}^{(k-1)} - t^{(k)}\right\|^2 = \left\|\widehat{t}^{(k-1)} - \Pi_{\mathcal{T}}\left(t^{(k-1)} - \eta_t \cdot \widehat{\nabla}_t\widehat{\mathcal{L}}(\pi_{\theta^\dagger}, t^{(k-1)}, \lambda^{(k-1)})\right)\right\|^2
$$
$$
\leq \left\|\widehat{t}^{(k-1)} - t^{(k-1)} + \eta_t \cdot \widehat{\nabla}_t\widehat{\mathcal{L}}(\pi_{\theta^\dagger}, t^{(k-1)}, \lambda^{(k-1)})\right\|^2
$$
$$
\text{(projection is non-expansive)}
$$
$$
= \left\|\widehat{t}^{(k-1)} - t^{(k-1)}\right\|^2 + \eta_t^2 \cdot \left\|\widehat{\nabla}_t\widehat{\mathcal{L}}(\pi_{\theta^\dagger}, t^{(k-1)}, \lambda^{(k-1)})\right\|^2
$$
$$
+ 2\eta_t\left\langle\widehat{t}^{(k-1)} - t^{(k-1)}, \widehat{\nabla}_t\widehat{\mathcal{L}}(\pi_{\theta^\dagger}, t^{(k-1)}, \lambda^{(k-1)})\right\rangle.
$$

Taking expectations, conditioned on $(t_{k-1}, \lambda_{k-1})$, and using Assumption 3.6 we have

$$
\mathbb{E}\left[\left\|\widehat{t}^{(k-1)} - t^{(k)}\right\|^2 \mid (t^{(k-1)}, \lambda^{(k-1)})\right]
$$
$$
\leq \mathbb{E}\left[\left\|\widehat{t}^{(k-1)} - t^{(k-1)}\right\|^2 \mid (t^{(k-1)}, \lambda^{(k-1)})\right]
$$
$$
+ 2\eta_t \cdot \left\langle t^{(k-1)} - \widehat{t}^{(k-1)}, \nabla_t f(t^{(k-1)}, \lambda^{(k-1)}) - b(\theta^*, \theta^\dagger, t^{(k-1)}, \lambda^{(k-1)})\right\rangle
$$
$$
+ 3\eta_t^2\left\|b_1(\theta^*, \theta^\dagger, t^{(k-1)}, \lambda^{(k-1)})\right\|^2 + 3\eta_t^2\left\|\nabla_t f(t^{(k-1)}, \lambda^{(k-1)})\right\|^2
$$
$$
+ 3\eta_t^2\mathbb{E}\left[\left\|\widehat{\nabla}_t\widehat{\mathcal{L}}(\pi_{\theta^*}, t^{(k-1)}, \lambda^{(k-1)}) + \nabla f(t^{(k-1)}, \lambda^{(k-1)})\right\|^2 \mid (t^{(k-1)}, \lambda^{(k-1)})\right].
$$
$$
\text{(using } \|x + y + z\|^2 \leq 3\|x\|^2 + 3\|y\|^2 + 3\|z\|^2)
$$

Taking expectations of both sides and using Lemma A.2 of [25], that $f(\cdot, \lambda)$ is $C$-Lipschitz (cf. Lemma 3.7), and Assumption 3.6,

$$\mathbb{E}\left\|\widehat{t}^{(k-1)} - t^{(k)}\right\|^2 \leq \mathbb{E}\left\|\widehat{t}^{(k-1)} - t^{(k-1)}\right\|^2$$
$$+ 2\eta_t \left\langle \widehat{t}^{(k-1)} - t^{(k-1)}, \nabla f(t^{(k-1)}, \lambda^{(k-1)}) - b(\theta^*, \theta^\dagger, t^{(k-1)}, \lambda^{(k-1)})\right\rangle$$
$$+ 3\eta_t^2(C^2 + \sigma^2 + \delta^2).$$

Since $\mathcal{T}$ is compact, define $\mathrm{diam}(\mathcal{T}) := \max_{x,y\in\mathcal{T}} \|x - y\|$. Using Cauchy-Schwarz on the term

$$2\eta_t \left\langle t^{(k-1)} - \widehat{t}^{(k-1)}, b_1(\theta^*, \theta^\dagger, t^{(k-1)}, \lambda^{(k-1)})\right\rangle$$
$$\leq 2\eta_t \left\|\widehat{t}^{(k-1)} - t^{(k-1)}\right\| \cdot \left\|b_1(\theta^*, \theta^\dagger, t^{(k-1)}, \lambda^{(k-1)})\right\|,$$

and substituting, yields

$$\mathbb{E}\left\|\widehat{t}^{(k-1)} - t^{(k)}\right\|^2 \leq \mathbb{E}\left\|\widehat{t}^{(k-1)} - t^{(k-1)}\right\|^2 + 2\eta_t \left\langle \widehat{t}^{(k-1)} - t^{(k-1)}, \nabla f(t^{(k-1)}, \lambda^{(k-1)})\right\rangle$$
$$+ 2\eta_t \delta \cdot \mathrm{diam}(\mathcal{T}) + 3\eta_t^2(C^2 + \sigma^2 + \delta^2). \tag{24}$$

From [25, Eq. (20)], we obtain

$$\Phi_{1/2\ell}(t^{(k)}) \leq \Phi_{1/2\ell}(t^{(k-1)}) + \ell \left\|\widehat{t}^{(k-1)} - t^{(k)}\right\|^2 \tag{25}$$

while, from [25, Enq. (23), (24)], we obtain

$$\left\langle \widehat{t}^{(k-1)} - t^{(k-1)}, \nabla_t f(t^{(k-1)}, \lambda^{(k-1)})\right\rangle$$
$$\leq f(\widehat{t}^{(k-1)}, \lambda^{(k-1)}) - f(t^{(k-1)}, \lambda^{(k-1)}) + \frac{\ell}{2}\left\|\widehat{t}^{(k-1)} - t^{(k-1)}\right\|^2 \tag{26}$$
$$f(\widehat{t}^{(k-1)}, \lambda^{(k-1)}) - f(t^{(k-1)}, \lambda^{(k-1)})$$
$$\leq \Phi(\widehat{t}^{(k-1)}) - f(t^{(k-1)}, \lambda^{(k-1)}) \leq \Delta^{(k-1)} - \frac{\ell}{2}\left\|\widehat{t}^{(k-1)} - t^{(k-1)}\right\|^2. \tag{27}$$

Plugging in (24) to (25), then combining (26), (27), and the fact that $\left\|\widehat{t}^{(k-1)} - t^{(k-1)}\right\| = \left\|\nabla\Phi_{1/2\ell}(t^{(k-1)})\right\|/2\ell$, finally gives

$$\mathbb{E}\left[\Phi_{1/2\ell}(t^{(k)})\right] \leq \mathbb{E}\left[\Phi_{1/2\ell}(t^{(k-1)})\right] + 2\eta_t\ell\Delta^{(k-1)} - \frac{\eta_t}{4}\mathbb{E}\left\|\nabla\Phi_{1/2\ell}(t^{(k-1)})\right\|^2$$
$$+ 2\eta_t\delta\ell \cdot \mathrm{diam}(\mathcal{T}) + 3\eta_t^2\ell(C^2 + \sigma^2 + \delta^2),$$

which completes the proof. □

**Lemma A.4.** *Let $\Delta^{(k)} = \mathbb{E}\left[\Phi(t^{(k)}) - f(t^{(k)}, \lambda^{(k)})\right]$ and $\lambda^*(t) \in \arg\max_{\lambda\in\Lambda} f(t, \lambda)$. The following holds for all $s \leq k - 1$:*

$$\Delta^{(k-1)} \leq 2\eta_t C\sqrt{C^2 + \sigma^2 + \delta^2}(2t - 2s - 1)$$
$$+ \frac{1}{2\eta_\lambda}\left(\mathbb{E}\left\|\lambda^{(k-1)} - \lambda^*(t^{(s)})\right\|^2 - \mathbb{E}\left\|\lambda^{(k)} - \lambda^*(t^{(s)})\right\|^2\right)$$
$$+ \mathbb{E}\left[f(t^{(k)}, \lambda^{(k)}) - f(t^{(k-1)}, \lambda^{(k-1)})\right] + \delta \cdot \mathrm{diam}(\Lambda) + 2\eta_\lambda(\sigma^2 + \delta^2).$$

*Proof.* For any $\lambda \in \Lambda$, the update of $\lambda^{(k)}$ and convexity of $\Lambda$ imply that

$$(\lambda - \lambda^{(k)})^\top\left(\lambda^{(k)} - \lambda^{(k-1)} - \eta_\lambda\nabla_\lambda f(t^{(k-1)}, \lambda^{(k-1)})\right) \geq 0.$$

Then, we have

$$\left\|\lambda - \lambda^{(k)}\right\|^2 \leq 2\eta_\lambda(\lambda^{(k-1)} - \lambda)^\top\nabla_\lambda f(t^{(k-1)}, \lambda^{(k-1)})$$

$$+ 2\eta_\lambda (\lambda^{(k)} - \lambda^{(k-1)})^\top \nabla_\lambda f(t^{(k-1)}, \lambda^{(k-1)})$$

$$+ \|\lambda - \lambda^{(k-1)}\|^2 - \|\lambda^{(k)} - \lambda^{(k-1)}\|^2$$

$$\leq 2\eta_\lambda (\lambda^{(k-1)} - \lambda^{(k)})^\top \widehat{\nabla}_\lambda \widehat{\mathcal{L}}(\pi_{\theta^\dagger}, t^{(k-1)}, \lambda^{(k-1)})$$

$$+ 2\eta_\lambda (\lambda^{(k-1)} - \lambda)^\top \nabla_\lambda f(t^{(k-1)}, \lambda^{(k-1)})$$

$$+ 2\eta_\lambda (\lambda^{(k)} - \lambda^{(k-1)})^\top \left( \nabla_\lambda f(t^{(k-1)}, \lambda^{(k-1)}) + \widehat{\nabla}_\lambda \widehat{\mathcal{L}}(\pi_{\theta^\dagger}, t^{(k-1)}, \lambda^{(k-1)}) \right)$$

$$+ \|\lambda - \lambda^{(k-1)}\|^2 - \|\lambda^{(k)} - \lambda^{(k-1)}\|^2,$$

where we let $\pi_{\theta^\dagger} = \pi_{\theta^\dagger(t^{(k-1)}, \lambda^{(k-1)})}$ (satisfying Assumption 3.6). Using Young's inequality,

$$\eta_\lambda (\lambda^{(k)} - \lambda^{(k-1)})^\top \left( \nabla_\lambda f(t^{(k-1)}, \lambda^{(k-1)}) + \widehat{\nabla}_\lambda \widehat{\mathcal{L}}(\pi_{\theta^\dagger}, t^{(k-1)}, \lambda^{(k-1)}) \right)$$

$$\leq \frac{1}{4} \left\| \lambda^{(k)} - \lambda^{(k-1)} \right\|^2 + \eta_\lambda^2 \left\| \widehat{\nabla}_\lambda \widehat{\mathcal{L}}(\pi_{\theta^\dagger}, t^{(k-1)}, \lambda^{(k-1)}) + \nabla_\lambda f(t^{(k-1)}, \lambda^{(k-1)}) \right\|^2 .$$

Taking expectations on both sides, conditioned on $(t^{(k-1)}, \lambda^{(k-1)})$, gives

$$\mathbb{E}\left[ \left\| \lambda - \lambda^{(k)} \right\|^2 \mid (t^{(k-1)}, \lambda^{(k-1)}) \right]$$

$$\leq 2\eta_\lambda \left( \lambda^{(k)} - \lambda^{(k-1)} \right)^\top \left( -\nabla_\lambda f(t^{(k-1)}, \lambda^{(k-1)}) + b_2(\theta^*, \theta^\dagger, t^{(k-1)}, \lambda^{(k-1)}) \right)$$

$$\text{(Assumption 3.6)}$$

$$+ 2\eta_\lambda \mathbb{E}\left[ \left( \lambda^{(k-1)} - \lambda \right)^\top \nabla_\lambda f(t^{(k-1)}, \lambda^{(k-1)}) \,\middle|\, (t^{(k-1)}, \lambda^{(k-1)}) \right]$$

$$+ \tfrac{1}{2} \mathbb{E}\left[ \left\| \lambda^{(k)} - \lambda^{(k-1)} \right\|^2 \,\middle|\, (t^{(k-1)}, \lambda^{(k-1)}) \right]$$

$$+ 2\eta_\lambda^2 \mathbb{E}\left[ \left\| \widehat{\nabla}_\lambda \widehat{\mathcal{L}}(\pi_{\theta^\dagger}, t^{(k-1)}, \lambda^{(k-1)}) + \nabla_\lambda f(t^{(k-1)}, \lambda^{(k-1)}) \right\|^2 \,\middle|\, (t^{(k-1)}, \lambda^{(k-1)}) \right]$$

$$+ \mathbb{E}\left[ \left\| \lambda - \lambda^{(k-1)} \right\|^2 \,\middle|\, (t^{(k-1)}, \lambda^{(k-1)}) \right] - \mathbb{E}\left[ \left\| \lambda^{(k)} - \lambda^{(k-1)} \right\|^2 \,\middle|\, (t^{(k-1)}, \lambda^{(k-1)}) \right].$$

Taking expectation on both sides (and Cauchy-Schwarz),

$$\mathbb{E} \left\| \lambda - \lambda^{(k)} \right\|^2$$

$$\leq 2\eta_\lambda \mathbb{E}\left[ \left( \lambda^{(k-1)} - \lambda^{(k)} \right)^\top \nabla_\lambda f(t^{(k-1)}, \lambda^{(k-1)}) + \left( \lambda^{(k-1)} - \lambda \right)^\top \nabla_\lambda f(t^{(k-1)}, \lambda^{(k-1)}) \right]$$

$$+ \mathbb{E} \left\| \lambda - \lambda^{(k-1)} \right\|^2 - \frac{1}{2} \mathbb{E} \left\| \lambda^{(k)} - \lambda^{(k-1)} \right\|^2 + 2\eta_\lambda \left\| \lambda^{(k-1)} - \lambda^{(k)} \right\| \cdot \left\| b_2(t^{(k-1)}, \lambda^{(k-1)}) \right\|$$

$$+ 2\eta_\lambda^2 \mathbb{E} \left\| \widehat{\nabla}_\lambda \widehat{\mathcal{L}}(\pi_{\theta^\dagger}, t^{(k-1)}, \lambda^{(k-1)}) + \nabla_\lambda f(t^{(k-1)}, \lambda^{(k-1)}) \right\|^2 .$$

From Lemma A.2, [25, Lemma A.2], and since $\mathrm{diam}(\Lambda)$ is finite, we get

$$\mathbb{E} \left\| \lambda - \lambda^{(k)} \right\|^2$$

$$\leq 2\eta_\lambda \mathbb{E}\left[ \left( \lambda^{(k-1)} - \lambda^{(k)} \right)^\top \nabla_\lambda f(t^{(k-1)}, \lambda^{(k-1)}) + \left( \lambda^{(k-1)} - \lambda \right)^\top \nabla_\lambda f(t^{(k-1)}, \lambda^{(k-1)}) \right]$$

$$+ \mathbb{E} \left\| \lambda - \lambda^{(k-1)} \right\|^2 - \frac{1}{2} \mathbb{E} \| \lambda^{(k)} - \lambda^{(k-1)} \|^2 + 2\eta_\lambda \delta \cdot \mathrm{diam}(\Lambda) + 4\eta_\lambda^2 (\sigma^2 + \delta^2) .$$

Since $f(t_{k-1}, \cdot)$ is concave and $\Lambda$ is convex (take $\eta_\lambda \leq 1/2\ell$),

$$\mathbb{E} \left\| \lambda - \lambda^{(k)} \right\|^2 \leq \mathbb{E} \left\| \lambda - \lambda^{(k-1)} \right\|^2 + 2\eta_\lambda (f(t^{(k-1)}, \lambda^{(k)}) - f(t^{(k-1)}, \lambda))$$

$$+ 2\eta_\lambda \delta \cdot \mathrm{diam}(\Lambda) + 4\eta_\lambda^2(\sigma^2 + \delta^2)\,.$$

Substituting $\lambda = \lambda^*(t^{(s)})$ (where $s \leq k - 1$),

$$\mathbb{E}\left[f(t^{(k-1)}, \lambda^*(t^{(s)})) - f(t^{(k-1)}, \lambda^{(k)})\right]$$
$$\leq \frac{1}{2\eta_\lambda}\left(\mathbb{E}\left\|\lambda^{(k-1)} - \lambda^*(t^{(s)})\right\|^2 - \mathbb{E}\left\|\lambda^{(k)} - \lambda^*(t^{(s)})\right\|^2\right) + \delta \cdot \mathrm{diam}(\Lambda) + 2\eta_\lambda(\sigma^2 + \delta^2)\,.$$

By the definition of $\Delta^{(k-1)}$,

$$\Delta^{(k-1)} \leq \mathbb{E}\Bigg[f(t^{(k-1)}, \lambda^*(t^{(k-1)})) - f(t^{(k-1)}, \lambda^*(t^{(s)}))$$
$$+ \left(f(t^{(k)}, \lambda^{(k)}) - f(t^{(k-1)}, \lambda^{(k-1)})\right) + \left(f(t^{(k-1)}, \lambda^{(k)}) - f(t^{(k)}, \lambda^{(k)})\right)\Bigg]$$
$$+ \frac{1}{2\eta_\lambda}\left(\mathbb{E}\left\|\lambda^{(k-1)} - \lambda^*(t^{(s)})\right\|^2 - \mathbb{E}\left\|\lambda^{(k)} - \lambda^*(t^{(s)})\right\|^2\right)$$
$$+ \delta \cdot \mathrm{diam}(\Lambda) + 2\eta_\lambda(\sigma^2 + \delta^2)\,.$$

Following the steps in [25, Lemma D.4], using that $f(\cdot, \lambda)$ is $C$-Lipschitz (by Lemma 3.7) and Lemma A.2, we have

$$\mathbb{E}\left[f(t^{(k-1)}, \lambda^*(t^{(k-1)})) - f(t^{(s)}, \lambda^*(t^{(k-1)}))\right] \leq 2\eta_t C\sqrt{C^2 + \sigma^2 + \delta^2}(t - 1 - s)$$
$$\mathbb{E}\left[f(t^{(s)}, \lambda^*(t^{(s)})) - f(t^{(k-1)}, \lambda^*(t^{(s)}))\right] \leq 2\eta_t C\sqrt{C^2 + \sigma^2 + \delta^2}(t - 1 - s)$$
$$\mathbb{E}\left[f(t^{(k-1)}, \lambda^{(k)}) - f(t^{(k)}, \lambda^{(k)})\right] \leq 2\eta_t C\sqrt{C^2 + \sigma^2 + \delta^2}\,.$$

Using [25, Eqn. (25)],

$$\Delta^{(k-1)} \leq 2\eta_t C\sqrt{C^2 + \sigma^2 + \delta^2} \cdot (2t - 2s - 1)$$
$$+ \frac{1}{2\eta_\lambda}\left(\mathbb{E}\left\|\lambda^{(k-1)} - \lambda^*(t^{(s)})\right\|^2 - \mathbb{E}\left\|\lambda^{(k)} - \lambda^*(t^{(s)})\right\|^2\right)$$
$$+ \mathbb{E}\left[f(t^{(k)}, \lambda^{(k)}) - f(t^{(k-1)}, \lambda^{(k-1)})\right] + \delta \cdot \mathrm{diam}(\Lambda) + 2\eta_\lambda(\sigma^2 + \delta^2)\,.$$

$\square$

**Lemma A.5.** *Let* $\Delta^{(k)} = \mathbb{E}\left[\Phi(t^{(k)}) - f(t^{(k)}, \lambda^{(k)})\right]$. *Let* $B \leq J + 1$ *be such that* $(J + 1)/B$ *is an integer. The following holds*

$$\frac{1}{J+1}\left(\sum_{k=0}^{J} \Delta^{(k)}\right) \leq 2\eta_t C\sqrt{C^2 + \sigma^2 + \delta^2}(B + 1)$$
$$+ \frac{\mathrm{diam}(\Lambda)^2}{2B\eta_\lambda} + \delta \cdot \mathrm{diam}(\Lambda) + 2\eta_\lambda(\sigma^2 + \delta^2) + \frac{\widehat{\Delta}_0}{J+1}\,.$$

*Proof.* We divide $\left\{\Delta^{(k)}\right\}_{k=0}^{J}$ into blocks where each block contains at most $B$ terms:

$$\left\{\Delta^{(k)}\right\}_{k=0}^{B-1}, \left\{\Delta^{(k)}\right\}_{k=B}^{2B-1}, \ldots, \left\{\Delta^{(k)}\right\}_{k=J-B+1}^{J}\,.$$

Then,

$$\frac{1}{J+1}\left(\sum_{k=0}^{J} \Delta^{(k)}\right) \leq \frac{B}{J+1}\left[\sum_{i=0}^{(J+1)/B-1}\left(\frac{1}{B}\sum_{k=iB}^{(i+1)B-1} \Delta^{(k)}\right)\right]\,. \tag{28}$$

Letting $s = 0$ and applying Lemma A.4,

$$\sum_{k=0}^{B-1} \Delta^{(k)} \leq 2\eta_t C \sqrt{C^2 + \sigma^2 + \delta^2}\, B^2 + \frac{1}{2\eta_\lambda} \mathbb{E} \left\| \lambda^{(0)} - \lambda^*(t^{(0)}) \right\|^2$$

$$+ \mathbb{E}\left[ f(t^{(B)}, \lambda^{(B)}) - f(t^{(0)}, \lambda^{(0)}) \right] + \delta B \cdot \mathrm{diam}(\Lambda) + 2\eta_\lambda B(\sigma^2 + \delta^2)$$

$$\leq 2\eta_t C \sqrt{C^2 + \sigma^2 + \delta^2}\, B^2 + \mathbb{E}\left[ f(t^{(B)}, \lambda^{(B)}) - f(t^{(0)}, \lambda^{(0)}) \right]$$

$$+ \frac{\mathrm{diam}(\Lambda)^2}{2\eta_\lambda} + \delta B \cdot \mathrm{diam}(\Lambda) + 2\eta_\lambda B(\sigma^2 + \delta^2)\,.$$

Letting $s = iB$ and applying Lemma A.4,

$$\sum_{k=iB}^{(i+1)B-1} \Delta^{(k)} \leq 2\eta_t C \sqrt{C^2 + \sigma^2 + \delta^2} B^2 + \mathbb{E}\left[ f(t^{(iB+B)}, \lambda^{(iB+B)}) - f(t^{(iB)}, \lambda^{(iB)}) \right]$$

$$+ \frac{\mathrm{diam}(\Lambda)^2}{2\eta_\lambda} + \delta B \cdot \mathrm{diam}(\Lambda) + 2\eta_\lambda B(\sigma^2 + \delta^2)\,.$$

Substituting these into (28):

$$\frac{1}{J+1}\left( \sum_{k=0}^{J} \Delta^{(k)} \right) \leq 2\eta_t C \sqrt{C^2 + \sigma^2 + \delta^2}\, B + \frac{1}{J+1}\mathbb{E}\left[ f(t_{J+1}, \lambda_{J+1}) - f(t_0, \lambda_0) \right]$$

$$+ \frac{\mathrm{diam}(\Lambda)^2}{2B\eta_\lambda} + \delta \cdot \mathrm{diam}(\Lambda) + 2\eta_\lambda(\sigma^2 + \delta^2)\,.$$

By the Lipschitzness of $f(\cdot, \lambda)$,

$$f(t_{J+1}, \lambda_{J+1}) - f(t_0, \lambda_0) \leq \eta_t C^2 (J+1) + \widehat{\Delta}_0\,.$$

This yields

$$\frac{1}{J+1}\left( \sum_{k=0}^{J} \Delta_k \right) \leq 2\eta_t C \sqrt{C^2 + \sigma^2 + \delta^2}\,(B+1)$$

$$+ \frac{\mathrm{diam}(\Lambda)^2}{2B\eta_\lambda} + \delta \cdot \mathrm{diam}(\Lambda) + 2\eta_\lambda(\sigma^2 + \delta^2) + \frac{\widehat{\Delta}_0}{J+1}\,.$$

$\square$

**Proof of Theorem 3.12**

Summing up the inequality from Lemma A.3, over $k = 1, \ldots, J+1$, yields

$$\mathbb{E}\left[ \Phi_{1/2\ell}(t^{(J+1)}) \right] \leq \Phi_{1/2\ell}(t^{(0)}) + 2\eta_t \ell \left( \sum_{k=0}^{J} \Delta^{(k)} \right) - \frac{\eta_t}{4}\left( \sum_{k=0}^{J} \mathbb{E}\left\| \nabla\Phi_{1/2\ell}(t^{(k)}) \right\|^2 \right)$$

$$+ \left( 2\eta_t \delta \ell \cdot \mathrm{diam}(\mathcal{T}) + 3\eta_t^2 \ell(C^2 + \sigma^2 + \delta^2) \right)(J+1)\,.$$

Applying Lemma A.5,

$$\mathbb{E}\left[ \Phi_{1/2\ell}(t^{(J+1)}) \right] \leq \Phi_{1/2\ell}(t^{(0)})$$

$$+ 2\eta_t \ell(J+1)\Bigg( 2\eta_t C \sqrt{C^2 + \sigma^2 + \delta^2}(B+1)$$

$$+ \frac{\mathrm{diam}(\Lambda)^2}{2B\eta_\lambda} + \delta \cdot \mathrm{diam}(\Lambda) + 2\eta_\lambda(\sigma^2 + \delta^2) \Bigg)$$

$$+ 2\eta_t \ell \widehat{\Delta}_0 - \frac{\eta_t}{4} \left( \sum_{k=0}^{J} \mathbb{E} \left\| \nabla \Phi_{1/2\ell}(t^{(k)}) \right\|^2 \right)$$

$$+ \left( 2\eta_t \delta \ell \cdot \operatorname{diam}(\mathcal{T}) + 3\eta_t^2 \ell (C^2 + \sigma^2 + \delta^2) \right)(J+1).$$

By the definition of $\widehat{\Delta}_\Phi$, we obtain

$$\frac{1}{J+1} \left( \sum_{k=0}^{J} \mathbb{E} \left\| \nabla \Phi_{1/2\ell}(t^{(k)}) \right\|^2 \right)$$

$$\leq \frac{4\widehat{\Delta}_\Phi}{\eta_t (J+1)} + 8\ell \left( 2\eta_t C \sqrt{C^2 + \sigma^2 + \delta^2}(B+1) + \frac{\operatorname{diam}(\Lambda)^2}{2B\eta_\lambda} + \delta \operatorname{diam}(\Lambda) + 2\eta_\lambda(\sigma^2 + \delta^2) \right)$$

$$+ \frac{8\ell \widehat{\Delta}_0}{J+1} + 12\eta_t \ell (C^2 + \sigma^2 + \delta^2) + 8\ell \delta \cdot \operatorname{diam}(\mathcal{T}).$$

Letting $B = \frac{\operatorname{diam}(\Lambda)}{2} \sqrt{\frac{1}{\eta_t \eta_\lambda C \sqrt{C^2 + \sigma^2 + \delta^2}}}$, we obtain

$$\frac{1}{J+1} \left( \sum_{k=0}^{J} \mathbb{E} \left\| \nabla \Phi_{1/2\ell}(t^{(k)}) \right\|^2 \right) \leq \frac{4\widehat{\Delta}_\Phi}{\eta_t (J+1)} + 24\ell \operatorname{diam}(\Lambda) \sqrt{\frac{\eta_t C \sqrt{C^2 + \sigma^2 + \delta^2}}{\eta_\lambda}}$$

$$+ 16\eta_\lambda \ell (\sigma^2 + \delta^2) + \frac{8\ell \widehat{\Delta}_0}{J+1}$$

$$+ 12\eta_t \ell (C^2 + \sigma^2 + \delta^2) + 8\ell \delta \cdot (\operatorname{diam}(\mathcal{T}) + \operatorname{diam}(\Lambda)).$$

With the choice of step sizes

$$\eta_\lambda = \min \left\{ \frac{1}{2\ell}, \frac{\epsilon^2}{16\ell(\sigma^2 + \delta^2)} \right\}, \tag{29}$$

$$\eta_t = \min \left\{ \begin{array}{l} \mathcal{O}\left( \dfrac{\epsilon^2}{\ell \left( C^2 + \sigma^2 + \delta^2 \right)} \right), \\[1.5em] \mathcal{O}\left( \dfrac{\epsilon^4}{\ell^3 \operatorname{diam}(\Lambda)^2} C \sqrt{C^2 + \sigma^2 + \delta^2} \right), \\[1.5em] \mathcal{O}\left( \dfrac{\epsilon^6}{\ell^3 \operatorname{diam}(\Lambda)^2 \left( \sigma^2 + \delta^2 \right) C \sqrt{C^2 + \sigma^2 + \delta^2}} \right) \end{array} \right\}, \tag{30}$$

we have

$$\frac{1}{J+1} \left( \sum_{k=0}^{J} \mathbb{E} \left\| \nabla \Phi_{1/2\ell}(t_k) \right\|^2 \right)$$

$$\leq \frac{4\widehat{\Delta}_\Phi}{\eta_t (J+1)} + \frac{8\ell \widehat{\Delta}_0}{J+1} + 8\ell \delta \cdot (\operatorname{diam}(\mathcal{T}) + \operatorname{diam}(\Lambda)) + \mathcal{O}(\epsilon^2).$$

Thus, we have an iteration complexity of

$$\mathcal{O}\left( \left( \frac{\ell(C^2 + \sigma^2 + \delta^2) \cdot \widehat{\Delta}_\Phi}{\epsilon^4} + \frac{\ell \widehat{\Delta}_0}{\epsilon^2} \right) \cdot \max \left\{ 1, \frac{\ell^2 \operatorname{diam}(\Lambda)^2}{\epsilon^2}, \frac{\ell^2 \operatorname{diam}(\Lambda)^2 (\sigma^2 + \delta^2)}{\epsilon^4} \right\} \right)$$

for recovering an $\mathcal{O}(\sqrt{\epsilon^2 + \delta \ell (\operatorname{diam}(\mathcal{T}) + \operatorname{diam}(\Lambda))})$-stationary point (cf. Definition 3.11). By simplifying, we obtain the desired iteration bound:

$$\mathcal{O}\left( \left( \frac{\ell^3 (C^2 + \sigma^2 + \delta^2)(\operatorname{diam}(\Lambda))^2 \cdot \widehat{\Delta}_\Phi}{\epsilon^6} + \frac{\ell^3 (\operatorname{diam}(\Lambda))^2 \cdot \widehat{\Delta}_0}{\epsilon^4} \right) \max \left\{ 1, \frac{\sigma^2 + \delta^2}{\epsilon^2} \right\} \right).$$

$\square$

# B  Experimental Details

## B.1  Simulations

Constraint tasks are a central class of real-world RL problems with applications in robotics, autonomous vehicles, and industrial control. To simulate realistic conditions, we use the locomotion tasks from the Safety-Gymnasium suite [19]. Further details, including visualizations of both constrained and unconstrained agent behaviors, are available on their website[6].

**Velocity Cost**  The velocity cost is defined as:

$$\text{cost} = \text{bool}(\text{vel}_{\text{current}} > \text{vel}_{\text{threshold}}),$$

where the velocity thresholds $\text{vel}_{\text{threshold}}$ for the tested environments are listed in Table 3. As reported by Towers et al. [47], these thresholds are set to 50% of the maximum velocity achieved by each agent after PPO training for $10^7$ steps. At each time step, the agent's instantaneous velocity is computed as:

$$\text{vel}_{\text{current}} = \sqrt{\text{vel}_{x,\text{current}}^2 + \text{vel}_{y,\text{current}}^2}, \tag{31}$$

where $\text{vel}_{x,\text{current}}$ and $\text{vel}_{y,\text{current}}$ are the agent's instantaneous velocities along the x- and y-axes, respectively, as provided by the simulator.

**Environments**  We use the latest version (v1) of Safety-Gymnasium without modifying the state, action, or reward space. Constraint violations do not alter the agent's behavior or environment dynamics. As a result, vanilla PPO converges to high reward solutions with large cumulative costs—effectively ignoring the constraints. All actions are normalized to the range $[-1, 1]$. To isolate the effects of our method, we reset the agent to the same initial state after every termination—a common practice in robotics.

**Episode Terminations**  Episodes terminate either when the time limit is reached or when the agent fails (e.g., by falling). Notably, HalfCheetah, Swimmer, and all safe navigation environments have no failure condition; their episodes always end due to the time limit.

Table 3: Environment-specific parameters used in our experiments.

| Variable | Safe Navigation | HalfCheetah | Hopper | Swimmer | Walker2d |
|---|---|---|---|---|---|
| Threshold $c$ | 0.0 | 3.2096 | 0.7402 | 0.2282 | 2.3415 |
| Failure condition | ✗ | ✗ | ✓ | ✗ | ✓ |
| Time limit (steps) | 500 | 1000 | 1000 | 1000 | 1000 |
| Initial CVaR variable $t_{\text{init}}$ | 0.0 | -1.3 | -0.1 | -0.0 | -0.975 |

**Stochasticity for Risk Management**  The environment is fully deterministic—identical actions from the same state always yield the same rewards and transitions—making it difficult to evaluate risk-sensitive behavior. To simulate uncertainty without altering the environment's internal dynamics, we inject zero-mean Gaussian noise (std. 0.05, i.e., 5% of the action range) into *all* agent actions at every step during both training and evaluation. This controlled perturbation introduces stochasticity in action execution, enabling us to assess how well the agent manages risk under uncertain conditions while maintaining consistent environment behavior.

**Evaluation**  Agents are evaluated every 1000 time steps by averaging the undiscounted sum of rewards over 10 episodes. The PPO agent uses the mean action, ensuring consistency in evaluation. Evaluations are entirely separate from training—no data is stored, and no network updates are performed.

---

[6]https://safety-gymnasium.readthedocs.io/en/latest/environments/safe_velocity.html

## B.2 Proximal Policy Optimization – *solver*

We use Proximal Policy Optimization [39] as a *solver* to learn a policy (i.e., line 4 in Algorithm 1). Our method serves as a *wrapper*, described next, that modifies the agent's raw reward to incorporate risk measures and constraints.

PPO first collects rollouts of state-action-reward sequences using the current policy, storing them as trajectories. Once sufficient data is gathered, it applies minibatch learning, splitting the rollout data into smaller batches and iteratively updating the policy over multiple epochs. During this process, the dual and CVaR variables remain fixed.

**Neural Networks** The value function and policy are approximated by neural networks, each with two hidden layers of 64 neurons using the `tanh` activation function. The value network processes states $s$ and outputs a scalar value. The policy network takes states $s$ as input, extracts hidden features, and passes them to a Gaussian distribution with learnable mean and standard deviation parameters. The action $a$ is then sampled from this distribution.

Table 4: PPO hyperparameters used in the experiments.

| Hyperparameter | Value |
|---|---|
| Optimizer | Adam |
| Learning rate (all networks) | $3 \times 10^{-4}$ |
| Linear learning rate decay | ✓ |
| Adam $\epsilon$ | $10^{-6}$ |
| Adam $\alpha$ | 0.99 |
| # rollout steps | 2048 |
| # minibatches per rollout | 32 |
| # epochs | 10 |
| Discount factor $\gamma$ | 0.99 |
| GAE $\lambda$ | 0.95 |
| Entropy coefficient | 0.0 |
| Value loss coefficient | 0.5 |
| Maximum gradient norm | 0.5 |
| Clip parameter | 0.2 |

**Hyperparameters** We employ Generalized Advantage Estimation (GAE) [38] to estimate advantages in PPO. The hyperparameters used by the PPO agent is provided in Table 4.

## B.3 Reward-Based SGD with Risk Constraints – *wrapper*

We follow the same rollout strategy to optimize the dual and CVaR variables. First, the policy is updated using collected rollouts while keeping $\lambda$ and $t$ fixed. Then, a new rollout is collected with the updated policy and used to update $\lambda$ and $t$, while keeping the policy parameters frozen.

### B.3.1 Implementation

To balance reward maximization with constraint handling, we frame the problem as standard risk-neutral reward maximization subject to a constraint that regulates violations through the conditional value-at-risk (CVaR) of the constraint quantity.

Let $r : \mathcal{S} \times \mathcal{A} \to \mathbb{R}$ be the reward function and $v : \mathcal{S} \times \mathcal{A} \to \mathbb{R}$ a constraint-quantifying function, e.g., velocity function in (31). We want to solve:

$$\sup_{\pi \in \mathcal{P}(\mathcal{S})} \mathbb{E}\left[\sum_{\tau=0}^{\infty} \gamma^{\tau} r(s_{\tau}, \pi(s_{\tau}))\right] \quad \text{s.t.} \quad \mathrm{CVaR}_{\nu\pi}^{\beta}(v(s,a)) \le c.$$

This constraint can be equivalently written as

$$\mathrm{CVaR}_{\nu\pi}^{\beta}(v(s,a)) \le c \iff -\mathrm{CVaR}_{\nu\pi}^{\beta}(-v(s,a)) \ge -c,$$

Table 5: Common hyperparameter values used across all environments.

| Variable | Value |
| --- | --- |
| VaR level $\beta$ | 0.3 |
| # trajectories used to compute gradients of $\lambda$ and $t$ | 8 |
| Initial dual variable $\lambda_{\text{init}}$ | 0.0 |
| Step size $\eta_\lambda$ | $5 \times 10^{-5}$ |
| Learning rate decay on $\eta_\lambda$ | ✗ |
| Step size $\eta_t$ | $5 \times 10^{-5}$ |
| Learning rate decay on $\eta_t$ | ✗ |

which aligns with the supremal convolution form of the reflected CVaR. Rewriting the constraint:

$$-\text{CVaR}_{\nu^\pi}^\beta(-\upsilon(s,a)) \geq -c \iff \sup_{t \in \mathbb{R}} \mathbb{E}\left[\sum_{\tau=0}^\infty \gamma^\tau \left(t - \frac{1}{\beta}(t + \upsilon(s_\tau, \pi(s_\tau)))_+\right)\right] \geq -\frac{c}{1-\gamma}.$$

Thus, we solve the following constrained problem:

$$\sup_{\pi \in \mathcal{P}(\mathcal{S}), t \in \mathbb{R}} \mathbb{E}\left[\sum_{\tau=0}^\infty \gamma^\tau r(s_\tau, \pi(s_\tau))\right]$$

$$\text{s.t.} \quad \mathbb{E}\left[\sum_{\tau=0}^\infty \gamma^\tau \left(t - \frac{1}{\beta}(t + \upsilon(s_\tau, \pi(s_\tau)))_+\right)\right] \geq -\frac{c}{1-\gamma}.$$

We implement this by modifying the reward at each time step using the Lagrangian:

$$r(s_\tau, \pi(s_\tau)) + \lambda_i \left(c + t - \frac{1}{\beta}(t + \upsilon(s_\tau, \pi(s_\tau)))_+\right), \tag{32}$$

where PPO is used as a black-box solver for the inner maximization over $\pi$, while $\lambda$ and $t$ are updated using single stochastic gradient descent and ascent steps as in Algorithm 1.

### B.3.2 Hyperparameters

All hyperparameters used in our algorithm are listed in Tables 3 and 5.

**Setting $\beta$**  The parameter $\beta$ controls which quantile mean is constrained. Higher values (close to 1) enforce more risk-neutral constraints, while lower values focus on rare events. We chose $\beta = 0.3$ to strike a balance: strong enough risk control to avoid constraint violations, while keeping the problem solvable.

**Number of Trajectories for Gradient Computation**  We tested $n = \{2, 4, 8, 16\}$ (in Algorithm 1) and concluded that $n = 8$ offers the best trade-off between runtime and gradient smoothness.

**Step Sizes $\eta_\lambda$ and $\eta_t$**  Step sizes were extensively tuned on Hopper and Walker2d. We tested values from $10^{-3}$ to $10^{-7}$. The best-performing configuration—$\eta_\lambda = \eta_t = 5 \times 10^{-5}$—was selected based on the convergence of $\lambda$ and $t$ within 15M time steps.

**Initial Value of $t$**  Since $t$ must take values in the negative real range (due to the supremal form), we expect it to converge to the negative value-at-risk. Thus, we initialized $t$ such that, with step size $\eta_t = 5 \times 10^{-5}$, its magnitude could reach the velocity threshold over training.

**Initial Value of $\lambda$**  Because $\lambda$ scales the penalty term added to the reward in (32), we initialized it neutrally with $\lambda = 0.0$ to avoid overly aggressive penalties at the start.

### B.4 Computational Resources

All experiments were performed on a computing system powered by an AMD Ryzen processor with 64 cores and 512 GB of RAM. A single NVIDIA RTX A6000 GPU with 48 GB VRAM was used for neural network training.

