# OpenReview forum: "Risk-Averse Constrained Reinforcement Learning with Optimized Certainty Equivalents"
_NeurIPS.cc/2025/Conference — NeurIPS 2025 poster_

### Official Review · Reviewer_uAkR · 2025-06-11

**Clarity:** 3
**Significance:** 3
**Originality:** 2
**Rating:** 4
**Confidence:** 3

**Summary:**

This work proposes a risk-aware constrained RL problem with both reward-based objective and constraints, for which this work established parameterized strong duality relation, online primal-dual algorithm and its convergence result, and verified the convergence by experimental results.

**Questions:**

(1) **My major concern** is in the setting novelty. "Our work is the first to handle reward-based constraints, covering a large class of risk measures (OCEs)" in the introduction looks like an overclaim, since [12, a] use reward-based constraints.

Your setting seems to be a special case of the risk measure-constrained RL (RCRL) [a], by restricting the function $g$ in Eq. (2) of [a] to $\{g(\cdot-t):t\in\mathbb{R}\}$ for a fixed $g$.

Could you cite [a] and describe your advantage over [a]?

Also, you present your results in CVAR and claim they hold for general OCEs at the beginning of Section 3.1. Can existing CVAR results like [12] also extend to general OCEs?

**I would like to raise my rating if you can solve this concern.**

[12] Yinlam Chow, Mohammad Ghavamzadeh, Lucas Janson, and Marco Pavone. Risk-constrained reinforcement learning with percentile risk criteria. Journal of Machine Learning Research, 18, 12 2015.

[a] Kim D, Cho T, Han S, et al. Spectral-Risk Safe Reinforcement Learning with Convergence Guarantees, Neurips 2024.

(2) To ensure Lemma 3.2 holds for general OCEs, what conditions should the function $g$ satisfy?

(3) Could you define $\widehat{\nabla} _ {t,\lambda}$ in Assumption 3.6? You could add the definition to your revised paper.

(4) How can Theorem 3.12 indicate the quality of the output policy?

(5) To make the experiments reproducible, could you explicitly define the state $s$, action $a$, reward function $r$ (in the objective)?

Minor issues:

(6) Abstract line 3: "through".

(7) You could describe the Slater's condition, possibly by an equation.

**Ethical Concerns:**

["NO or VERY MINOR ethics concerns only"]

**Final Justification:**

My questions including my major concern are well solved.
I also read the comments and rebuttal for reviewer uZsP who gave rejection rating, and feel that the rebuttal has also solved reviewer uZsP's comments.

Therefore, I raised my rating to 4.

**Limitations:**

Yes, the authors have written their limitations in the conclusion section.

**Quality:**

3

**Strengths And Weaknesses:**

Strengths: I can understand this work clearly. The risk-aware RL problem is popular and important for safe reinforcement learning. The algorithm and theoretical results are clear and reasonable based on my knowledge about minimax optimization and reinforcement learning.

Weaknesses (See my questions below for details): The major concern is that problem novelty is not as strong as claimed. The convergence result does not directly imply the quality of policy that we are interested in. Some points need clarification.

---

> ### Author Rebuttal · Authors · 2025-07-31
>
> Thank you for your detailed review and thoughtful comments. We hope the responses below address your questions clearly.
>
> ---
> ### **Questions**
>
> 1.	Thank you for the question! We originally included a table clarifying this distinction, but omitted it due to page constraints. We will take care to make this distinction clearer in the text. The key difference is that the constraint in [12] applies the risk measure to the discounted return of a cost function (see Eqs. (2) and (3), and the definition of $\mathcal{J}^\theta(x^0)$ at the bottom of p.5 in [12]). We refer to this as the return-based formulation, in contrast to the reward-based formulation used in our work. In our framework’s notation, the return-based formulation (like in [12, a]) can be written as:
>
> $$\max_{t} \left(t - \frac{1}{\beta} \left(t - \mathbb{E}\left[\sum_{\tau=0}^\infty \gamma^\tau r(s_\tau, a_\tau) \right]\right)_+ \right),$$
>
>
> while the reward-based formulation (our work), which places the risk measure inside the occupancy measure, becomes:
>
> $$\max_{t} \mathbb{E}\left[ \sum_{\tau=0}^\infty \gamma^\tau \left(t - \frac{1}{\beta}(t - r(s_\tau, a_\tau))_+ \right) \right].$$
>
> Similarly, in [a], the spectral risk measure is applied to $G_{C_i}^\pi$ (Eq. (5)), which is also a return (defined at the bottom of p.2). This places it within the return-based framework of [12], though extended to more general spectral risk measures.
>
> We hope this clarifies the distinction and will make these points more explicit in the manuscript. There is more that we can say and we are happy to further elaborate or follow up as needed.
>
>
> 2.	We will add this to the background section. The conditions on $g$ appear as Definition 2.1 in Ben-Tal and Teboulle [4]. Specifically, $g$ is a utility function that is proper, closed, concave, and non-decreasing, and it satisfies $g(0) = 0$ along with $1 \in \partial g(0)$, where $ \partial g(\cdot)$ denotes the subdifferential map.
>
> 3.	As noted in our response to Reviewer WADh, we will make sure to include this explicitly. The subgradient can be computed using Equation (8) and the definition of $r'$ provided on line 149, and we will ensure that this is clearly stated in the manuscript.
>
> 4.	This is a good question! Like many nonconvex problems that arise in DL/RL, we can usually only guarantee convergence to different notions of local stationary points rather than true global optima. In our setting, we use the Moreau envelope which is standard in nonsmooth weakly convex optimization. As written in Theorem 3.12, the quality of the policy depends on the scale of $\delta$ to be small (we want Assumption 3.6 to say that the RL solver returns nearly the optimal policy for the inner problem) and we’d ideally want the non-convexity to be such that all $\epsilon$-stationary points have values close to the global optima. We could state Theorem 3.12 in the main body more simply (e.g., assuming $\delta = O(\epsilon^2)$) if it would help the presentation.
>
> 5.	The objective function—that is, the modified reward implementing the risk-aware constrained objective—is explicitly defined in terms of the state, action, and original reward function in Appendix B.3.1, Equation (25).
>
> The state and action spaces are defined by the joint positions, velocities, and other physical properties of the agents in the MuJoCo simulator. These elements vary across environments and are governed by the physics engine, as documented on the MuJoCo website. The Safety-Gymnasium environments we use are built directly on MuJoCo v4, and we do not modify the state, action, or reward definitions provided therein. Therefore, loading any environment from the suite automatically applies the default state-action-reward configuration.
>
> Given the complexity and high dimensionality of these environments, we were unable to include some of these details in the main body. However, we will point readers to the simulator's official documentation and, if the paper is accepted, elevate the content of Appendix B.3.1 to the main body using the additional page allowance.
>
> 6.	Thank you! We now fixed it.
>
> 7.	We will define Slater’s condition in the Appendix and reference it accordingly. Slater’s condition, a strict feasibility constraint, is often necessary to establish strong duality. It states that there exists a policy which is strictly feasible (i.e., there exists some policy which satisfies the constraints with strict inequality).

---

> ### Author Response · Authors · 2025-08-06
> **Additional Experiments**
>
> ## Safe Navigation Results Added
>
> Thank you for acknowledging our responses and for your participation in the rebuttal period. We would like to follow up with additional experiments conducted during the rebuttal phase. These new experiments were performed on the _Safe Navigation_ tasks in the Safety-Gymnasium suite.
>
> ---
> The table below summarizes the converged episodic cumulative reward and cost, denoted by $R$ and $C$, respectively, for our algorithm and the unconstrained vanilla PPO.
>
> | **Algo/Env** |        PointCircle1             | PointGoal1               | PointPush1              |
> |:--------------------------|:------------------------:|:------------------------:|:-----------------------:|
> | PPO                       | $R=60.18$, $C=206.74$    | $R=21.89$, $C=45.09$     | $R=0.93$, $C=38.48$     |
> | Ours                      | $R=39.19$, $C=0.0$       | $R=13.56$, $C=0.0$       | $R=2.42$, $C=0.0$       |
>
> - Our method is able to effectively minimize cost while maintaining high rewards, consistent with our findings in the Safety-Velocity experiments—**even in the presence of discrete constraints**.
>
> - Notably, in the Push environment, our algorithm achieves **higher rewards** than vanilla PPO while **maintaining zero cost**. This is due to the fact that the vanilla PPO agent occasionally gets stuck on hazards and fails to reach the goal—suggesting that it struggles to find a global optimum in the absence of constraints and our algorithm works like a _regularizer of the path_.
>
> - As expected from our Safety-Velocity experiments, reducing cost generally results in some reward degradation—highlighting the inherent trade-off in risk-constrained optimization.
>
> ---
> We will incorporate these results into the final version if the paper is accepted. Once again, thank you for your constructive comments—they have helped improve the quality of our work.

---

### Official Review · Reviewer_enYi · 2025-06-30

**Clarity:** 2
**Significance:** 3
**Originality:** 3
**Rating:** 4
**Confidence:** 2

**Summary:**

This paper proposes a risk-aware constrained reinforcement learning framework that incorporates reward-based risk measures into the objective and constraints. It extends prior work by enabling constrained settings and establishes a strong duality result that allows for an exact partial Lagrangian relaxation. The authors develop a modular online algorithm compatible with various RL methods and provide theoretical convergence guarantees. Experiments on standard benchmarks demonstrate improved risk management and reduced constraint violations.

**Questions:**

While the proposed method is theoretically distinguished from Bonetti et al. [7], particularly in its ability to handle constrained settings and inexact policy solvers, the lack of experimental comparison with [7] leaves some ambiguity regarding its practical advantages. Given that the Safe Velocity benchmark does not enforce hard constraints on dynamics (i.e., constraint violations do not terminate episodes), it may be feasible to apply [7] in this setting, at least in a limited form. Even a simple empirical comparison—e.g., reward, CVaR, or policy stability—would significantly enhance the reader’s understanding of how the proposed method performs relative to prior work. If direct implementation is difficult, an ablation study or visualization highlighting the behavioral differences would still provide valuable insight into the practical impact of the proposed approach.

**Ethical Concerns:**

["NO or VERY MINOR ethics concerns only"]

**Final Justification:**

Since additional experiments with the baseline methods have not been included at this point, I will maintain my original score. The strengths and weaknesses remain as previously stated.

**Limitations:**

yes

**Quality:**

3

**Strengths And Weaknesses:**

- Strengths
  - The proposed method ensures an exact equivalence to the original constrained problem via parameterized strong Lagrangian duality, providing solid theoretical guarantees.
  - The modular algorithm can be easily integrated with standard RL solvers such as PPO, making it practical and broadly applicable.
  - The proposed method addresses tail risks and catastrophic events, which are critical in high-stakes applications.
  - The evaluation is structured around three measurable goals—convergence of auxiliary variable to CVaR-beta, stabilization of the dual variable \lambda, and safe yet reward-maximizing policy behavior, which align well with the theoretical contributions.
  - The authors simulate realistic uncertainty by injecting Gaussian noise into agent actions in an otherwise deterministic environment, enabling a meaningful evaluation of risk-sensitive learning.
- Weaknesses
  - The experiments are limited to a single benchmark suite (Safety-Gymnasium), which restricts the ability to assess the proposed method's generalizability to other domains or task types.
  - The paper lacks a comparative evaluation against baseline methods. While the authors provide a clear theoretical distinction from Bonetti et al. [7]—noting that [7] is limited to the unconstrained setting and assumes exact policy solvers—the practical impact of these differences remains unclear. Since both methods operate in similar reward-based risk-aware frameworks, a direct empirical comparison (even if limited) would strengthen the justification for the proposed approach.

---

> ### Author Rebuttal · Authors · 2025-07-31
>
> Thank you for your thoughtful review and for raising this question, which we had considered during the development of the manuscript. We hope the following response effectively addresses the identified weaknesses and provides a clear answer to your question.
>
> ---
> ### **Weaknesses**
>
> 1. We primarily selected Safety-Velocity tasks as they are based on MuJoCo environments, which are widely recognized benchmarks in deep RL. These tasks are also the most realistic and complex within the Safety-Gymnasium suite, featuring high-dimensional state and action spaces, 3D spatial structure, and intricate physics dynamics. In contrast, many of the other tasks in the suite are either toy problems, Atari-style environments, or limited to 2D settings.
>
> Nonetheless, we have initiated experiments on Safety-Navigation as well and will include the results if they are completed in time for a camera-ready version.
>
> 2. Please refer to our response to your question below.
>
> ---
> ### **Question**
>
> Thank you for the thoughtful question! Our experiments were primarily designed to demonstrate the theoretical properties and practical behavior of our algorithm, rather than to benchmark against prior methods. That said, we acknowledge the value of such comparisons.
>
> We were unable to implement the approach of Bonetti et al. [7], as their block-coordinate descent algorithm as written in the paper is missing implementation detail and is not accompanied by publicly available code. (Our attempts to obtain it from the authors were also unsuccessful.) Nonetheless, our framework can be adapted to approximate their setting—for example, by fixing the Lagrange multiplier to zero and disabling updates—allowing us to mimic an unconstrained optimization scenario similar to [7]. If the reviewer believes such a comparison would improve the clarity or impact of our presentation, we are happy to include it.
>
> We also note that our approach, which leverages stochastic gradient methods, is conceptually distinct from the optimization scheme in [7]. Even in the absence of a direct implementation, we will consider including ablations or visualizations to highlight behavioral differences, should space and time permit.

---

> > ### Comment · Reviewer_enYi · 2025-08-04
> >
> > Thank you for answering my question. I appreciate your reply about the comparison with the baseline. The proposed experiments should be added. Since the experiment results are not currently available, I will keep my score.

---

> ### Author Response · Authors · 2025-08-06
> **Experiments on Safety Navigation**
>
> ## New Experiments Added
>
> We appreciate your continued engagement during the discussion period. Following your comments, we began experiments on SafetyNavigation, which have been completed now. Below, we provide the experimental details and results in numerical form, in accordance with rebuttal guidelines that prohibit the use of external links, images, or plots.
>
>
> ---
> ### Experimental Details
>
> We have added three new tasks from the Navigation section of Safety-Gymnasium:
> - **Circle:** The agent must move in a circular trajectory around the center while staying within boundaries. Exiting the circular area from the inside results in a cost of +1. Two walls are created by constraining the agent along the x-axis.
> - **Goal:** The agent must navigate to the goal while avoiding 8 hazards. Each contact with a hazard incurs a cost of +1, with a maximum total cost of 8.
> - **Push:** The agent must push a box to the goal location while avoiding 2 hazards. Each contact with a hazard incurs a cost of +1, with a maximum total cost of 2.
>
> To manage time constraints, we used the point agent (with 12-dimensional state and 2-dimensional action space), since the constraint optimization task is independent of the agent’s body. Using a more complex agent would primarily affect the PPO solver’s efficiency, not our wrapper algorithm. This setup differs from Safety-Velocity, where the agent’s physical design (e.g., joints) directly impacts constraint violations such as velocity.
>
> As in our original experiments, we introduced action noise to simulate uncertainty and highlight the risk-awareness capabilities of our algorithm.
>
> **Note:** Constraints in these navigation tasks are inherently discrete, in contrast to the continuous constraint shaping in Safety-Velocity. As a result, dual and CVaR variable learning is less smooth, making the **optimization problem inherently more challenging**.
>
> ---
> ### Results
>
> The table below summarizes the converged episodic cumulative reward and cost, denoted by $R$ and $C$, respectively, for our algorithm and the unconstrained vanilla PPO. The vanilla PPO results are consistent with those reported in Table 5 of the Safety-Gymnasium paper.
>
> | **Algo/Env** |        PointCircle1             | PointGoal1               | PointPush1              |
> |:--------------------------|:------------------------:|:------------------------:|:-----------------------:|
> | PPO                       | $R=60.18$, $C=206.74$    | $R=21.89$, $C=45.09$     | $R=0.93$, $C=38.48$     |
> | Ours                      | $R=39.19$, $C=0.0$       | $R=13.56$, $C=0.0$       | $R=2.42$, $C=0.0$       |
>
>
> - Our method is able to effectively minimize cost while maintaining high rewards, consistent with our findings in the Safety-Velocity experiments—**even in the presence of discrete constraints**.
>
> - Notably, in the Push environment, our algorithm achieves **higher rewards** than vanilla PPO while **maintaining zero cost**. This is due to the fact that the vanilla PPO agent occasionally gets stuck on hazards and fails to reach the goal—suggesting that it struggles to find a global optimum in the absence of constraints and our algorithm works like a _regularizer of the path_.
>
> - As expected from our Safety-Velocity experiments, reducing cost generally results in some reward degradation—highlighting the inherent trade-off in risk-constrained optimization.
>
> ---
> ### Conclusion
>
>
> We will include these results, along with relevant discussion, in the camera-ready version if the paper is accepted.
>
> That said, we would be grateful if you would consider raising your score. We welcome any additional suggestions or feedback.

---

### Official Review · Reviewer_6DSm · 2025-06-30

**Clarity:** 3
**Significance:** 2
**Originality:** 2
**Rating:** 4
**Confidence:** 3

**Summary:**

This work proposes a risk-averse constrained reinforcement learning framework that integrates Optimized Certainty Equivalents (OCEs) to address the limitation of traditional constrained RL, which neglects tail risks in high-stakes applications. By formulating objectives and constraints using reward-based risk measures, the framework achieves per-stage robustness in both reward values and time. It establishes a parameterized strong Lagrangian duality, ensuring exact equivalence to the original constrained problem under appropriate constraint qualifications. The authors design a modular algorithm that can wrap standard RL solvers and prove the convergence of the algorithm under common assumptions. Experimental results on locomotion tasks in Safety-Gymnasium demonstrate that the method can effectively reduce violations of risk constraints while maintaining competitive reward performance, validating its risk-aware properties.

**Questions:**

1.The constraint (Assumption 3.4) is crucial for precise duality, yet its practical effectiveness across different environments remains unclear. Could the authors provide empirical evidence of violations of this assumption and discuss the algorithm's performance in such scenarios?

2.This paper focuses on CVaR, but OCE encompasses other risk measures (e.g., entropy-regularized variants). How does the framework perform under these measures? Supplementary comparisons would highlight its flexibility and identify which OCEs are most effective for constrained reinforcement learning.

3.The experiments exclude high-dimensional environments such as Ant. Could the authors test the method in such environments to verify its scalability? It would also be valuable to supplement comparative experiments with existing state-of-the-art baseline algorithms.

4.How sensitive is the algorithm to β across different environments? Could sensitivity analysis be supplemented to clarify its impact on the risk-return trade-off?

**Ethical Concerns:**

["NO or VERY MINOR ethics concerns only"]

**Limitations:**

yes

**Quality:**

3

**Strengths And Weaknesses:**

Strengths：

1. This work is technically rigorous with a solid theoretical foundation, providing detailed proofs of parameterized strong duality and non-asymptotic convergence guarantees, along with basic simulation experiments. The authors candidly acknowledge the limitations of this paper and validate the relevant claims with consistent results.

Weaknesses：

1. The constraint conditions for exact duality (Assumption 3.4) are relatively stringent, and their practical applicability across different environments has not been thoroughly examined. Furthermore, this paper fails to provide sufficient comparative experiments.

2. The core ideas of this research (including OCE in reinforcement learning, Lagrangian relaxation of constraints, and CVaR in risk-aware optimization) are derived from existing concepts. Its originality lies more in integration rather than the proposal of entirely new concepts, which significantly undermines the innovativeness of this research.

3. The connection between the partial Lagrangian relaxation method and the original constrained problem lacks the necessary intuitive explanation, which impairs the readability of the paper.

---

> ### Author Rebuttal · Authors · 2025-07-31
>
> Thank you for your review and insightful comments. We hope the responses below effectively address the identified weaknesses and answer your questions.
>
> ---
> ### **Weaknesses**
>
> 1. Please see our response to Question 1 below.
>
> 2. Thank you for your feedback. While our work builds on established tools, we believe the contributions lie in their nontrivial integration and the resulting theoretical framework. The conclusions we draw and the guarantees we establish are not straightforward applications of existing results but require careful analysis and novel synthesis across these domains.
>
> 3.	We will revise the exposition to make this connection clearer. Specifically, the problem in Eq. (6) corresponds to the left-hand side of the equation on line 180. We will add a reference to the earlier constrained problem and briefly remind the reader of the Lagrangian duality framework to improve clarity and readability.
>
> ---
> ### **Questions**
>
> 1. This is a valid point—thank you for raising it. Although this condition may be hard to check in practice, the framework still serves as a valid Lagrangian relaxation—even when this assumption is not met and in the absence of strong duality—as is often the case in constrained optimization.
>
> 2. We focus on CVaR as it is one of the most widely used and well-understood risk measures, which we believe helps with clarity of exposition. That said, our framework is designed to accommodate a broad class of OCEs. We can add explicit equations and explanations in the Appendix to illustrate how the formulation and algorithm would adapt to other OCEs to further highlight the flexibility of our approach.
>
> 3. Thank you for the suggestion! We will aim to include additional experiments, including comparisons with state-of-the-art baselines, if the paper is accepted. As noted in the limitations section, we anticipate that computational cost may become a bottleneck in high-dimensional settings.
>
> In our preliminary experiments, we evaluated a vanilla PPO agent in a deterministic, non-constrained Ant environment. Across five seeds, the agent failed to converge to meaningful rewards, indicating that the solver was insufficient for this setting. Consequently, we excluded Ant from our main experiments.
>
> We also ran vanilla PPO on the reported environments as a baseline for comparison. However, to better isolate the practical effectiveness of our proposed risk-aware method, we chose not to report those results. The vanilla PPO agent violated constraints in 85–90% of episode steps, and the rewards were highly volatile due to noise injected into the actions.
>
> If preferred, we can include the vanilla PPO results in the final version. We will also add a brief note in Appendix B.1 explaining our findings in the Ant environment.
>
> 4. While the algorithm itself is not directly sensitive to the $\beta$ parameter, the constraint-adhering behavior of the final policy—i.e., the average frequency of constraint violations—is. By definition, $\beta$ specifies the risk level by setting the percentile of the worst-case outcomes over which the expected cost is computed. For instance, if we use a larger $\beta$, and assuming the dual and CVaR variables converge, we would generally observe more constraint violations, as the policy is optimized over a broader range of outcomes and is less focused on avoiding the worst-case scenarios.

---

> > ### Comment · Reviewer_6DSm · 2025-08-04
> >
> > Thank you very much for your careful revisions and detailed responses to my comments. Most of my previous concerns have been addressed or clarified, and I appreciate the effort you’ve put into refining the work. I still encourage you to further validate your proposed algorithm in a broader range of experimental scenarios.

---

> ### Author Response · Authors · 2025-08-06
> **Experiments on Safety Navigation**
>
> ## New Experiments Added
>
> Thank you for your participation in the discussion period! We are glad that your concerns have been addressed. In response to your feedback, we carried out experiments on SafetyNavigation tasks, which are now complete. Below, we present the experimental details and results numerically, in accordance with rebuttal guidelines.
>
> ---
> ### Experimental Details
>
> We have added three new tasks from the Navigation section of Safety-Gymnasium:
> - **Circle:** The agent must move in a circular trajectory around the center while staying within boundaries. Exiting the circular area from the inside results in a cost of +1. Two walls are created by constraining the agent along the x-axis.
> - **Goal:** The agent must navigate to the goal while avoiding 8 hazards. Each contact with a hazard incurs a cost of +1, with a maximum total cost of 8.
> - **Push:** The agent must push a box to the goal location while avoiding 2 hazards. Each contact with a hazard incurs a cost of +1, with a maximum total cost of 2.
>
> To manage time constraints, we used the point agent (with 12-dimensional state and 2-dimensional action space), since the constraint optimization task is independent of the agent’s body. Using a more complex agent would primarily affect the PPO solver’s efficiency, not our wrapper algorithm. This setup differs from Safety-Velocity, where the agent’s physical design (e.g., joints) directly impacts constraint violations such as velocity.
>
> As in our original experiments, we introduced action noise to simulate uncertainty and highlight the risk-awareness capabilities of our algorithm.
>
> **Note:** Constraints in these navigation tasks are inherently discrete, in contrast to the continuous constraint shaping in Safety-Velocity. As a result, dual and CVaR variable learning is less smooth, making the **optimization problem inherently more challenging**.
>
> ---
> ### Results
>
> The table below summarizes the converged episodic cumulative reward and cost, denoted by $R$ and $C$, respectively, for our algorithm and the unconstrained vanilla PPO. The vanilla PPO results are consistent with those reported in Table 5 of the Safety-Gymnasium paper.
>
> | **Algo/Env** |        PointCircle1             | PointGoal1               | PointPush1              |
> |:--------------------------|:------------------------:|:------------------------:|:-----------------------:|
> | PPO                       | $R=60.18$, $C=206.74$    | $R=21.89$, $C=45.09$     | $R=0.93$, $C=38.48$     |
> | Ours                      | $R=39.19$, $C=0.0$       | $R=13.56$, $C=0.0$       | $R=2.42$, $C=0.0$       |
>
>
> - Our method is able to effectively minimize cost while maintaining high rewards, consistent with our findings in the Safety-Velocity experiments—**even in the presence of discrete constraints**.
>
> - Notably, in the Push environment, our algorithm achieves **higher rewards** than vanilla PPO while **maintaining zero cost**. This is due to the fact that the vanilla PPO agent occasionally gets stuck on hazards and fails to reach the goal—suggesting that it struggles to find a global optimum in the absence of constraints and our algorithm works like a _regularizer of the path_.
>
> - As expected from our Safety-Velocity experiments, reducing cost generally results in some reward degradation—highlighting the inherent trade-off in risk-constrained optimization.
>
> ---
> ### Bottom Line
> We will incorporate these results and the corresponding discussion into the camera-ready version, if the paper gets accepted.
>
> We would also appreciate it if you could consider updating your score. We remain open to any further suggestions or feedback you may have.

---

### Official Review · Reviewer_WADh · 2025-07-03

**Clarity:** 3
**Significance:** 3
**Originality:** 3
**Rating:** 5
**Confidence:** 3

**Summary:**

Many existing safe RL works rely on expected cumulative constraints, which are not applicable to safety-critical RL problems. To this end, this paper proposes a risk-aware constrained/safe RL scheme. Several theoretical guanrantees such as the convergence and the parameterized strong duality are established. In addition, the practical implementation can be wrapped around the standard unconstrained RL algorithms like PPO, SAC etc. The proposed method is also substantiated by numerical experiments conducted on well-known Safety-Gymnasium benchmarks.

**Questions:**

1. Does the stepsize need to be carefully tuned? How can we select a proper stepsize for a new/unknown environment?

2. The authors should include more references regarding probabilistic-constraints (a.k.a. chance-constraints) in safe RL.

3. The authors should provide more experiments to validate the robustness of the proposed method on other type of tasks, e.g., Safe Navigation, Safe Vision, etc. They are all provided in Safety-Gymnasium.

4. Please provide the exact formulations or details for the subgradients in Line 5 of Algorithm 1.

**Ethical Concerns:**

["NO or VERY MINOR ethics concerns only"]

**Final Justification:**

The rebuttal looks good given the limited time. I would encourage the authors to include more advanced safe RL baselines in Safe Navigation for comparison, at least in the camera-ready version. At this stage, I will maintain my positive score.

**Limitations:**

yes

**Paper Formatting Concerns:**

No major formatting issues

**Quality:**

3

**Strengths And Weaknesses:**

**Strengths**:

1. This paper is well-written and well-presented.

2. The topic of this paper (risk-averse safe/constrained RL) is very significant for the community and real-world applications.

3. The proposed method is supported by solid theoretical guarantees and experimental validations on well-known safe RL benchmarks.

4. This work can be wrapped around the standard unconstrained RL algorithms such as PPO, TRPO, SAC, thus easy to implement.

5. The convergence of rewards and dual variables are validated by empirical evidence.


**Weaknesses**:

1. This paper does **NOT** compare with any other risk-aware safe RL baseline methods. It is hard to evaluate the practical performance of this work without comparing with some baselines.

2. No comments/discussions on the assumptions are provided, see e.g., Assumption 3.4. The proper discussions on the assumptions are necessary although I understand the concerns for main text space.

3. The proposed method is only evaluated on one type of tasks, i.e., safe_velocity. Please see my suggestions later.

---

> ### Author Rebuttal · Authors · 2025-07-31
>
> Thank you for the careful review and thoughtful suggestions! We address the identified weaknesses and respond to your specific questions below.
>
> ---
> ### **Weaknesses**
>
> 1. Please see our responses below.
>
> 2.  We included some discussions of assumptions in Appendix A.6 and A.7 (for Assumptions 3.6 and 3.8, respectively). We will take care to add a brief discussion of Assumption 3.4 as well but in the main body to improve readability.
>
> 3. Please see our responses below.
>
> ---
> ### **Questions**
>
> 1.	We did not carefully or extensively tune any of the step sizes.
>
> - **PPO agent:** We did not experiment with alternative step sizes for the PPO solver. We adopted a value previously found effective in our other projects using the same MuJoCo environments.
>
> - **CVaR and Dual Variable:** For the CVaR and dual variables, we used a smaller learning rate relative to that of the networks (e.g., the policy), based on the common practice of using lower step sizes for scalar parameters due to their increased sensitivity to gradient noise and lack of averaging effects present in high-dimensional updates. We tested three magnitudes: $5 \times 10^{-4}$, $5 \times 10^{-5}$ (chosen), and $5 \times 10^{-6}$. The middle value provided better stability than the first and faster convergence than the last.
>
> 2.	We will add additional references and relevant discussion on probabilistic (chance) constraints in safe RL. If the reviewer has specific examples of related work they believe are particularly relevant to our approach, we would be happy to consider and include them.
>
> 3.	We primarily chose Safety-Velocity tasks because they are based on MuJoCo environments, which are widely used benchmarks in deep RL. These tasks are among the most realistic and challenging within the Safety-Gymnasium suite, involving high-dimensional state and action spaces, 3D spatial configurations, and complex physics dynamics. In contrast, many other tasks in the suite, such as those in Safe-Navigation and Safe-Vision, tend to be simpler—often limited to 2D or relying on lower-dimensional inputs.
>
> That said, in response to your suggestion, we have begun experiments on Safety-Navigation. We did not include Safety-Vision due to the additional need for CNN-based image processing, which may not be feasible within the current timeline. If completed in time, we will include Safety-Navigation results in the camera-ready version.
>
> 4.	Thank you for the suggestion! The subgradient in Line 5 of Algorithm 1 can be computed using Equation (8) along with the definition of $r'$ provided on line 149. Still, we will make sure to include this formulation explicitly in the paper for clarity.

---

> ### Author Response · Authors · 2025-08-06
> **Additional Experiments**
>
> ## Safe Navigation Results Added
>
> Thank you once again for your constructive suggestions and for recognizing the merits of our work. As you had encouraged, we conducted additional experiments on the SafetyNavigation tasks in Safety-Gymnasium.
>
> These experiments are now complete, and we would like to follow up by providing the details and numerical results below, in accordance with rebuttal guidelines that prohibit the use of external links, images, or plots.
>
>
>
> ---
> ### Experimental Details
>
> We have added three new tasks from the Navigation section of Safety-Gymnasium:
> - **Circle:** The agent must move in a circular trajectory around the center while staying within boundaries. Exiting the circular area from the inside results in a cost of +1. Two walls are created by constraining the agent along the x-axis.
> - **Goal:** The agent must navigate to the goal while avoiding 8 hazards. Each contact with a hazard incurs a cost of +1, with a maximum total cost of 8.
> - **Push:** The agent must push a box to the goal location while avoiding 2 hazards. Each contact with a hazard incurs a cost of +1, with a maximum total cost of 2.
>
> To manage time constraints, we used the point agent (with 12-dimensional state and 2-dimensional action space), since the constraint optimization task is independent of the agent’s body. Using a more complex agent would primarily affect the PPO solver’s efficiency, not our wrapper algorithm. This setup differs from Safety-Velocity, where the agent’s physical design (e.g., joints) directly impacts constraint violations such as velocity.
>
> As in our original experiments, we introduced action noise to simulate uncertainty and highlight the risk-awareness capabilities of our algorithm.
>
> **Note:** Constraints in these navigation tasks are inherently discrete, in contrast to the continuous constraint shaping in Safety-Velocity. As a result, dual and CVaR variable learning is less smooth, making the **optimization problem inherently more challenging**.
>
> ---
> ### Results
>
> The table below summarizes the converged episodic cumulative reward and cost, denoted by $R$ and $C$, respectively, for our algorithm and the unconstrained vanilla PPO. The vanilla PPO results are consistent with those reported in Table 5 of the Safety-Gymnasium paper.
>
> | **Algo/Env** |        PointCircle1             | PointGoal1               | PointPush1              |
> |:--------------------------|:------------------------:|:------------------------:|:-----------------------:|
> | PPO                       | $R=60.18$, $C=206.74$    | $R=21.89$, $C=45.09$     | $R=0.93$, $C=38.48$     |
> | Ours                      | $R=39.19$, $C=0.0$       | $R=13.56$, $C=0.0$       | $R=2.42$, $C=0.0$       |
>
>
> - Our method is able to effectively minimize cost while maintaining high rewards, consistent with our findings in the Safety-Velocity experiments—**even in the presence of discrete constraints**.
>
> - Notably, in the Push environment, our algorithm achieves **higher rewards** than vanilla PPO while **maintaining zero cost**. This is due to the fact that the vanilla PPO agent occasionally gets stuck on hazards and fails to reach the goal—suggesting that it struggles to find a global optimum in the absence of constraints and our algorithm works like a _regularizer of the path_.
>
> - As expected from our Safety-Velocity experiments, reducing cost generally results in some reward degradation—highlighting the inherent trade-off in risk-constrained optimization.
>
> ---
> We will include these results, along with the relevant discussion, in the camera-ready version if the paper is accepted.

---

> > ### Comment · Reviewer_WADh · 2025-08-08
> >
> > Thank you for the response and the additional experiments. They look good given the limited time in the rebuttal. I would encourage the authors to include more advanced safe RL baselines in Safe Navigation for comparison, at least in the camera-ready version. At this stage, I will maintain my positive score. Thank you!

---

### Official Review · Reviewer_uZsP · 2025-07-11

**Clarity:** 3
**Significance:** 3
**Originality:** 2
**Rating:** 3
**Confidence:** 3

**Summary:**

This paper proposes a novel framework for risk-averse constrained reinforcement learning (RL) by leveraging Optimized Certainty Equivalents (OCEs) to address the limitations of standard expectation-based RL formulations, which often fail to account for tail risks or catastrophic events. The core idea is to apply per-stage risk measures, rather than return-based ones—to both the objective and constraints using OCEs, such as Conditional Value-at-Risk (CVaR). They reformulate the constrained RL problem using a partial Lagrangian relaxation, which under suitable constraint qualifications yields an optimization problem that is provably equivalent to the original problem and amenable to a stochastic minimax solution. They show strong duality in this reformulation and provide a non-asymptotic convergence analysis of the resulting algorithm.
In the experiments, their algorithm wraps around any black-box RL solver such as PPO, enabling integration with existing pipelines. Experimental results demonstrate that the proposed method effectively manages risk while maintaining constraint satisfaction and stable convergence.

**Questions:**

Questions
- Please see the weaknesses section.
- The proposed method uses a standard black-box RL algorithm in the inner loop to optimize a modified reward function for fixed dual variables. It is well-known that the optimal policy for a CMDP is generally stochastic. However, many RL solvers, when optimizing for a fixed reward function, will output a deterministic optimal policy. This seems to create a potential mismatch: the outer-loop problem requires a search over the space of stochastic policies, while the inner-loop solver might only explore a subset of near-deterministic ones. Could the authors comment on this? Does this mismatch potentially prevent the algorithm from finding the true stochastic optimum of the original CMDP?
- Algorithm 1 specifies that the final policy is chosen by uniformly sampling an iterate j* from the entire training history. This seems counter-intuitive, as policies from early in the training (small j) are unlikely to satisfy the problem's constraints, given that the dual variables have not yet converged. Returning such a policy would be unacceptable in a safety-critical context. Could the authors clarify the rationale behind this uniform sampling strategy?

**Ethical Concerns:**

["NO or VERY MINOR ethics concerns only"]

**Final Justification:**

I appreciate the theoretical contributions of this work, and the theoretical rigor justifies raising the score. However, the lack of experimental comparison with established safe RL algorithms limits the impact of the work. Therefore, I am only raising the score to borderline reject. A more thorough experimental evaluation including comparisons to relevant safe RL baselines would be necessary for full acceptance.

**Limitations:**

.

**Paper Formatting Concerns:**

.

**Quality:**

3

**Strengths And Weaknesses:**

Strengths
- The paper addresses the crucial challenge of integrating risk-awareness into constrained RL, which is essential for deploying RL agents in many risk-averse problems in real-world.
- The paper presents a method to optimize the OCE-constrained objective for risk-averse reinforcement learning with theoretical analysis.


Weaknesses
- Insufficient comparison with baselines and related work. The paper fails to adequately position its contributions within the existing literature through direct methodological and empirical comparisons (e.g. [1,2,3]). This is a significant weakness as it makes it difficult to assess the practical advantages and novelty of the proposed framework.
- The paper's choice to constrain the per-step cost CVaR, rather than the more conventional return-based CVaR, is a major methodological decision that is not sufficiently justified. While this approach may be better suited for capturing instantaneous, catastrophic risks and offers theoretical tractability, it may also lead to overly conservative or myopic policies in scenarios where overall episodic safety is the primary goal. The paper lacks a discussion of this critical trade-off and fails to provide an empirical comparison against a return-based risk management baseline to demonstrate in which specific contexts its approach is superior. This omission makes it difficult to understand the true scope and limitations of the proposed framework.


[1] Chow et al., Risk-Sensitive and Robust Decision-Making: a CVaR Optimization Approach, 2015

[2] Ying et al., Towards Safe Reinforcement Learning via Constraining Conditional Value-at-Risk, 2022

[3] Achiam et al., Constrained Policy Optimization, 2017

---

> ### Author Rebuttal · Authors · 2025-07-31
>
> Thank you for your questions and observations! We hope that we are able to address the points of weakness and related questions below.
>
> ---
> ### **Weaknesses**
>
> 1. Thank you for pointing out these additional references. We will include these comparisons in the related work section.
>
>     - **Reference [1]:** We cite the later work of Chow et al. (reference [12] in our paper), which addresses constraints, whereas [1] focuses on replacing the expectation in the objective with CVaR in a return-based formulation without constraints.
>
>     - **Reference [2]:** It applies risk measures to constraints in the return-based setting and appears to focus on a risk-neutral objective with a CVaR constraint (see Eq. (11) in [2]).
>
>     - **Reference [3]:** It proposes solving CMDPs via a local policy search algorithm, where the constraint enforces proximity to the previous policy iterate using a distance metric. In contrast, our method is not based on local policy search and does not impose constraints based on policy distance.
>
>
> 2. Our goal is to position this work primarily as a theoretical contribution, with the following key strengths:
>     - a parameterized strong duality,
>     - an interesting interplay of convex analytic and Lagrangian duality,
>     - a modular algorithmic framework that accommodates a broad class of OCEs,
>     - and the ability to handle per-step cost CVaR constraints, which differs from most prior work that focuses on return-based CVaR.
>
> Notably, strong duality is a property that does not hold for many constrained problems, which are often solved via relaxation. We provide a complete theoretical treatment, including a novel convergence analysis and detailed discussion of assumptions and their practical relevance (see Appendices A.1, A.6, and A.7). In this regard, our approach aligns with prior theoretical works such as the given reference [1], which similarly emphasizes rigorous analysis over extensive empirical comparisons.
>
> That said, we acknowledge the importance of this trade-off and will aim to include additional experiments and return-based baselines, if time permits and the paper is accepted.
>
> ---
> ### **Questions**
>
> 1.	The PPO solver employs a stochastic policy, modeled as a Gaussian distribution with learnable mean and standard deviation, as noted on line 727. Importantly, our framework does not require the final policy to be deterministic. As you pointed out, recovering the true stochastic optimum of the original CMDP requires maintaining stochasticity, which our algorithm preserves through the use of such policies.
>
> We will add 1–2 clarifying sentences to the paragraph starting on line 281 to make this point explicit.
>
> 2.	In the min-max optimization literature, theoretical guarantees are typically established for average or randomly selected iterates, as last-iterate convergence is generally difficult to prove without additional structural assumptions. Uniform sampling of an iterate is a standard approach in nonconvex optimization, including for gradient descent and stochastic gradient descent, to ensure convergence to stationary points. In practice, one usually returns the iterate when the learning curve stops changing significantly.
>
> That said, in our practical implementation—given the use of a deep reinforcement learning solver in high-dimensional environments—we return the final policy observed when the learning curve stabilizes. This is noted on line 235 in our wrapper code (`wrapper > RA_C_RL.py`).
>
> Thank you for bringing this observation; we will clarify this point with 1–2 sentences in the text.

---

> ### Author Response · Authors · 2025-08-06
>
> ### **Choice of Per-Step Cost CVaR**
>  Further elaborating on the choice of per-step cost CVaR (as opposed to the return-based like most other works), we would like to point out that a general comparison of return-based and reward-based risk measures in the context of RL has been done in Bonetti et al. [7]. Our work explores the choice of using reward-based risk measures in both objective and constraints (or even a combination of risk-neutral and risk-aware) with theoretical guarantees, as solving constrained problems is more difficult (theoretically and empirically). We will take care to highlight these existing comparisons of the per-step cost risk measure and the return-based counterpart (citing Bonetti et al. [7]) in the text to include these critical trade-off discussions.
>
> ---
>
> ### **New Experiments Added**
> Moreover, we conducted additional experiments on the SafetyNavigation tasks in Safety-Gymnasium and compared them to a PPO baseline (which was the method of comparison for the return-based Chow et al. [12]). These experiments are now complete, and we provide the details and numerical results below, in compliance with rebuttal guidelines that restrict the use of external links, images, or plots.
>
> We have added three new tasks from the Navigation section of Safety-Gymnasium:
> - **Circle:** The agent must move in a circular trajectory around the center while staying within boundaries. Exiting the circular area from the inside results in a cost of +1. Two walls are created by constraining the agent along the x-axis.
> - **Goal:** The agent must navigate to the goal while avoiding 8 hazards. Each contact with a hazard incurs a cost of +1, with a maximum total cost of 8.
> - **Push:** The agent must push a box to the goal location while avoiding 2 hazards. Each contact with a hazard incurs a cost of +1, with a maximum total cost of 2.
>
> To manage time constraints, we used the point agent (with 12-dimensional state and 2-dimensional action space), since the constraint optimization task is independent of the agent’s body. Using a more complex agent would primarily affect the PPO solver’s efficiency, not our wrapper algorithm. This setup differs from Safety-Velocity, where the agent’s physical design (e.g., joints) directly impacts constraint violations such as velocity.
>
> As in our original experiments, we introduced action noise to simulate uncertainty and highlight the risk-awareness capabilities of our algorithm.
>
> **Note:** Constraints in these navigation tasks are inherently discrete, in contrast to the continuous constraint shaping in Safety-Velocity. As a result, dual and CVaR variable learning is less smooth, making the **optimization problem inherently more challenging**.
>
> ---
> ### Results
>
> The table below summarizes the converged episodic cumulative reward and cost, denoted by $R$ and $C$, respectively, for our algorithm and the unconstrained vanilla PPO. The vanilla PPO results are consistent with those reported in Table 5 of the Safety-Gymnasium paper.
>
> | **Algo/Env** |        PointCircle1             | PointGoal1               | PointPush1              |
> |:--------------------------|:------------------------:|:------------------------:|:-----------------------:|
> | PPO                       | $R=60.18$, $C=206.74$    | $R=21.89$, $C=45.09$     | $R=0.93$, $C=38.48$     |
> | Ours                      | $R=39.19$, $C=0.0$       | $R=13.56$, $C=0.0$       | $R=2.42$, $C=0.0$       |
>
> - Our method is able to effectively minimize cost while maintaining high rewards, consistent with our findings in the Safety-Velocity experiments—**even in the presence of discrete constraints**.
>
> - Notably, in the Push environment, our algorithm achieves **higher rewards** than vanilla PPO while **maintaining zero cost**. This is due to the fact that the vanilla PPO agent occasionally gets stuck on hazards and fails to reach the goal—suggesting that it struggles to find a global optimum in the absence of constraints and our algorithm works like a _regularizer of the path_.
>
> - As expected from our Safety-Velocity experiments, reducing cost generally results in some reward degradation—highlighting the inherent trade-off in risk-constrained optimization.
>
> We will include these results, along with the relevant discussion if the paper is accepted.

---

### Author Response · Authors · 2025-08-06
**Global Response**

We thank all reviewers for their constructive feedback and for recognizing the strengths of our work. Below, we summarize the common strengths and weaknesses raised across the reviews, along with our responses.

---
### Summary of the Responses

**Strengths:**
- The topic is significant and timely for the community (uZsP, WADh, uAkR)
- The manuscript is clearly written (WADh, uAkR)
- The paper is technically rigorous with a strong theoretical foundation (uZsP, WADh, 6DSm, enYi, uAkR)
- The modular algorithm can be integrated with standard RL solvers (WADh, enYi)
- The experimental results align well with the theoretical contributions (WADh, 6DSm, enYi)

**Weaknesses:**
- Limited empirical comparisons, such as reliance on a single benchmark suite or lack of comparison to baselines (uZsP, WADh, enYi)
  → **Response**: Please see the following section for newly added experiments addressing this point.

- Assumption 3.4 (Constraint Qualification) may be restrictive (WADh, 6DSm)
  → **Response**: While this condition may be difficult to verify in practice, the framework remains useful as a Lagrangian relaxation—even in the absence of strong duality—as is common in constrained optimization.

---
### New Experiments Added

In response to several reviewers, additional experiments were conducted to further validate the effectiveness of our approach, particularly in new environments. To that end, we conducted new experiments on the _Safe Navigation_ tasks from the Safety-Gymnasium suite.

Below is a summary of our new findings. We have also addressed each reviewer’s comments individually.

1. Added _3 new environments_, bringing the **total to 7**.
2. The new environments feature discrete constraints, unlike the continuous constraints in the original Safe Velocity experiments, making the **optimization problem inherently more challenging**.
3. Our algorithm continues to perform effectively—**maximizing rewards while maintaining zero cost**.

We will include these results, along with the corresponding discussion, in the camera-ready version if the paper is accepted. We will also add the unconstrained vanilla PPO results, which were completed prior to the initial submission.

---

### Note · Authors · 2025-08-11

We want to thank the reviewers again for their time, consideration, and engagement during this reviewing and rebuttal period. We have been able to improve the manuscript with the help of the various inputs and questions from reviewers. We appreciate the generally positive feedback regarding our work, and did our best to address all concerns and points of weakness. To the best of our knowledge, we have applied most of the changes we promised and are committed to making sure any remaining are done before a final version of the manuscript.

We have already summarized the common strengths and responded to common weaknesses from reviewers in the `Global Response` comment below, as well as responded to all reviewers individually. Therefore, we refrain from restating these in the final remarks and instead thank the reviewers (and chairs) again for their time and feedback.

---

### Decision · Program_Chairs · 2025-09-17

**Decision:**

Accept (poster)

**Comment:**

**Summary:** This paper investigated risk-averse constrained reinforcement learning (RL) using Optimized Certainty Equivalents (OCEs) as a unified risk-measure framework. It formulated constrained MDP objectives that (i) optimize an OCE-based performance criterion and/or (ii) enforce safety-style constraints (e.g., on costs or tail risks). This paper also derived policy-gradient estimators for OCE objectives and presented a primal–dual (Lagrangian) actor–critic scheme to handle constraints. Theoretical performance guarantees on convergence to  stationary points under some standard assumptions were provided. Finally, some experimental results were provided.


**Strength:**
- The problem itself is important and general. This paper brings a general risk-measure family, OCE to the constrained RL setting. Such a measure subsumes other popular choices like CVaR.
- The proposed methodologies are clearly presented, for instance, the derivation of OCE policy gradients and the primal-dual training loop are clean and likely reproducible in standard actor–critic codebases.
- This paper is guaranteed with strong theory. It provided convergence guarantees under some standard assumptions, and clarifies how constraint penalties and OCE parameters enter the updates.

**Weakness:**
- One key limitations or weakness pointed out by the reviewers is the limitations of the experimental evaluations. It is better to add some stronger risk-aware and constraint-satisfaction baselines.
- The theoretical analysis focuses on asymptotic/stationary-point convergence. There are no finite-sample or sample-complexity bounds, which are often provided in constrained RL literature.

**Reasons for acceptance:** This paper offers a general risk-averse formulation for constrained RL with a sound derivation and implementable algorithm. Theoretical analysis on the convergence was conducted and analyzed. The performance was also supported by experiments, including some additional experiments conducted and added during the rebuttals.

**Discussion \& rebuttal assessment:** The authors addressed most of the concerns raised by the reviewers during the rebuttal. A major concern was on the experimental evaluations. Given some additional experimental results provided during the rebuttals, the authors are highly encouraged to include them into the final version.